# PT-T2I/V: An Efficient Proxy-Tokenized Diffusion Transformer for Text-to-Image/Video-Task

**Jing Wang**[1,2*], **Ao Ma**[2*], **Jiasong Feng**[2*], **Dawei Leng**[2†], **Yuhui Yin**[2], **Xiaodan Liang**[1,3,4†]

[1]Shenzhen Campus of Sun Yat-Sen University, [2]360 AI Research, [3]Peng Cheng Laboratory,
[4]Guangdong Key Laboratory of Big Data Analysis and Processing,
`wangj977@mail2.sysu.edu.cn` `{maao,fengjiasong,lengdawei}@360.cn`
`xdliang328@gmail.com`

## Abstract

The global self-attention mechanism in diffusion transformers involves redundant computation due to the sparse and redundant nature of visual information, and the attention map of tokens within a spatial window shows significant similarity. To address this redundancy, we propose the **P**roxy-**T**okenized **D**iffusion **T**ransformer (**PT-DiT**), which employs sparse representative token attention (where the number of representative tokens is much smaller than the total number of tokens) to efficiently model global visual information. Specifically, within each transformer block, we compute an averaging token from each spatial-temporal window to serve as a proxy token for that region. The global semantics are captured through the self-attention of these proxy tokens and then injected into all latent tokens via cross-attention. Simultaneously, we introduce window and shift window attention to address the limitations in detail modeling caused by the sparse attention mechanism. Building on the well-designed PT-DiT, we further develop the PT-T2I/V family, which includes a variety of models for T2I, T2V, and T2MV tasks. Experimental results show that PT-DiT achieves competitive performance while reducing computational complexity in image and video generation tasks (e.g., a reduction 59% compared to DiT and a reduction 34% compared to PixArt-$\alpha$). The visual exhibition and code are available at `https://360cvgroup.github.io/Qihoo-T2X/`.

## 1 Introduction

Recent advancements in core diffusion models, including Sora (OpenAI, 2024), Kling (Huawei, 2024), Stable Diffusion 3 (Stability AI, 2024), PixArt-$\alpha/\Sigma/\delta$ (Chen et al., 2023; 2024a;b), Vidu (Shengshu AI, 2024), Lumina-T2X (Gao et al., 2024), Flux (BlackForestlabs AI, 2024), and CogVideoX (Yang et al., 2024), have led to significant achievements in the creation of photo-realistic image and video. Transformer-based models such as Sora and Vidu have demonstrated the ability to generate high-quality samples at arbitrary resolutions. These models also adhere strongly to scaling laws, achieving superior performance as parameter sizes increase. Additionally, Lumina-T2X has shown uniformity in performing various generation tasks, further validating the potential of the transformer-based architectures in diffusion models.

However, the quadratic complexity of global self-attention concerning sequence length increases the computational cost of the Diffusion Transformer, leading to practical challenges such as longer generation times and higher training costs. This issue also hinders the application of DiT to high-quality video generation. For example, while 3D attention-based approaches(Xu et al., 2024; Yang et al., 2024; Lab & etc., 2024; Gao et al., 2024) have demonstrated superiority over 2D spatial attention combined with 1D temporal attention counterparts(Zheng et al., 2024; Ma et al., 2024b; Bar-Tal et al., 2024; Blattmann et al., 2023; Lu et al., 2023), the extensive computational demands limit their scalability for higher-resolution and longer video generation. Current studies (Han et al., 2023; Koner et al., 2024; Yu et al., 2024) in visual understanding and recognition have highlighted that global attention mechanisms often exhibit redundancy due to the sparse and repetitive nature of visual information. Specifically, by visualizing the attention map, we observe that the attention of tokens within the same window is similar for spatially distant tokens, while differing for

---

*Equal Contribution. †Corresponding Authors

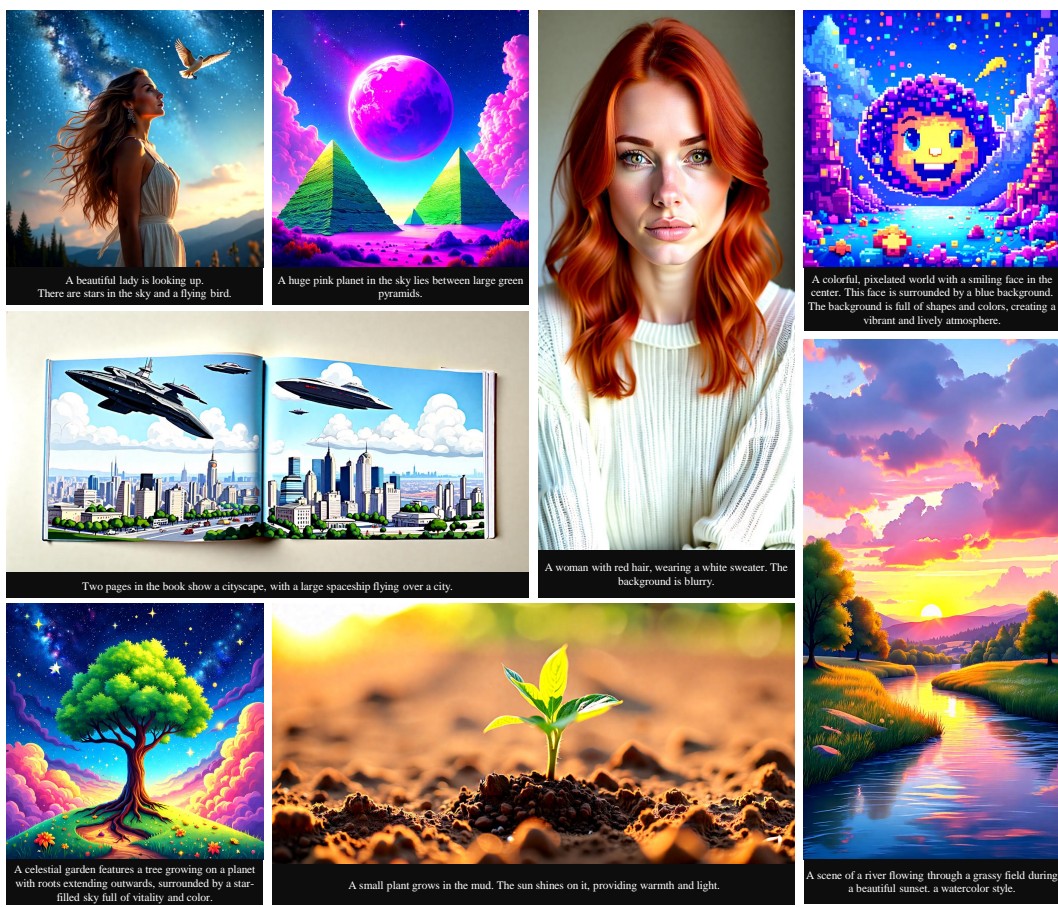

Figure 1: The samples from PT-T2I showcase high fidelity and aesthetic qualities, demonstrating a strong consistency with given textual descriptions.

spatially neighboring tokens, as illustrated in Fig. 3. This observation indicates that the dense long-range attention, which triggers significant computational overhead, is redundant. Thus, reducing this redundancy is believed to enhance the efficiency of Diffusion Transformers in generating higher-resolution images and longer videos.

In this paper, we propose the **P**roxy-**T**okenized **D**iffusion **T**ransformer (**PT-DiT**) and further present the **PT-T2I/V** series, which includes both Text-to-Image, Text-to-Video, and Text-to-MultiView generation models. To address the redundancy of visual information, PT-DiT employs proxy-tokenized attention instead of a global attention mechanism to reduce the computational complexity of visual token interaction. Specifically, we first recover the spatial and temporal relationships of the token sequence through a reshaping operation. Given the similarity of visual information within localized spatial regions and temporal frames, we calculate an averaging token from each spatial-temporal window as a representative token, forming a set of proxy tokens. The interaction and broadcasting of visual global information are then achieved through self-attention among proxy tokens and cross-attention between proxy tokens and all latent tokens. Additionally, to enhance the texture modeling capabilities, we introduce window attention and incorporate shift-window attention, similar to Swin Transformer (Liu et al., 2021), to avoid lattice artifacts as shown in Fig. 8.

With the well-designed proxy-tokenized attention, PT-DiT can be adapted to both image and video generation tasks without structural adjustments. For image generation, as shown in Fig. 2, compared to PixArt-$\alpha$ (Chen et al., 2023), our method achieves an approximate 33% reduction in computational complexity GFLOPs under the same parameter scale. For video generation, in contrast to 2D spatial and 1D temporal attention, which has limited spatial-temporal modeling, and 3D full-attention, which suffers from high computational complexity, PT-DiT can efficiently and comprehensively extracts 3D information, benefiting from proxy token interaction mechanisms.

Experimental results demonstrate that our method achieves competitive performance with significant efficiency. As shown in Fig. 1, PT-T2I can generate high-quality and high-fidelity images while closely adhering to the provided text instructions. Meanwhile, for the image generation task, PT-DiT's computational complexity is 51% of DiT and 66% of PixArt-$\alpha$ for the same parameter size. For the video generation task, despite having 3 million more parameters than EasyAnimateV4, the PT-DiT/H's computational complexity is only 82%

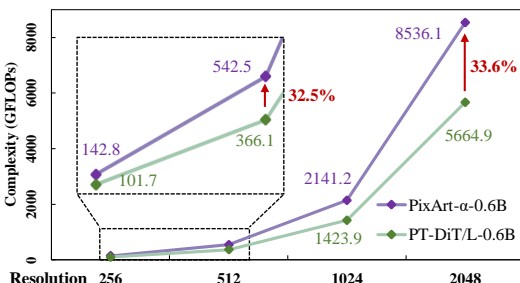

Figure 2: Comparison of complexity between PixArt-$\alpha$ and PT-DiT/L at various resolutions.

of EasyAnimateV4 (Xu et al., 2024) and 77% of CogVideoX (Yang et al., 2024) for the same parameter size. Overall, using the standard 3D VAE settings ($8\times$ spatial downsampling rate and $4\times$ temporal downsampling rate), experimental tests indicate that we can train the PT-DiT/XL (1.1B) model for images at a resolution of $2048 \times 2048$ or for video at a resolution of $512 \times 512 \times 288$ on the 64GB Ascend 910B (Huawei, 2024).

In summary, the unique contributions of this paper can be summarized as follows:

- We analyze redundant computations in self-attention within Diffusion Transformers, caused by visual sparsity and redundancy. We find that redundancy primarily arises in attention interactions within the same spatial window and between distant tokens, while modeling relationships between spatially adjacent tokens is crucial and should be preserved.

- We introduce a proxy token mechanism that leverages spatial priors for local token fusion to generate proxy tokens, and uses Proxy Token Attention and Visual Cross-Attention to efficiently establish and propagate global associations, while retaining all latent tokens to preserve detailed texture. Additionally, window and shifted window attention are employed to enhance the modeling of spatial proximity.

- Through extensive experimentation, we demonstrate that our efficient proxy-token-based diffusion transformer model achieves competitive performance with state-of-the-art T2I and T2V models. Our approach lays the foundation for T2I and T2V models across diverse scenarios, offering significant computational efficiency advantages. We will open-source both our models and code to support the advancement of efficient diffusion transformers.

## 2 RELATED WORK

**Image Generation with Diffusion Transformer.** Please refer to **Appendix.** A.1

**Video Generation with Diffusion Transformer.** Building on the advancements in image generation with Diffusion Transformers, recent work (Xu et al., 2024; Yang et al., 2024; Lab & etc., 2024; Zheng et al., 2024; Ma et al., 2024b; Lu et al., 2023) has been devoted to extending the DiT structure to video generation. EasyAnimateV2 (Xu et al., 2024), Open-Sora (Zheng et al., 2024) is based on PixArt-$\alpha$ (Chen et al., 2023), incorporating temporal 1D attention and utilizing a 3D VAE to generate additional frames. CogVideoX (Yang et al., 2024), Open-Sora-Plan (Lab & etc., 2024), Lumina-T2X (Gao et al., 2024) and EasyAnimateV4 (Xu et al., 2024) points out the shortcomings of temporal 1D attention and employs an expert transformer with 3D attention. While 3D attention effectively manages significant motion between adjacent frames, it also incurs a substantial computational cost. To overcome this challenge, we propose PT-DiT, which introduces an innovative token compression strategy. This strategy compresses not only in spatial dimensions but also across frames, enabling 3D attention with significantly reduced computational overhead.

**Efficient Diffusion Transformer.** As the application of transformers becomes more mature, there are some solutions proposed in different fields targeting the attention computation problem (Han et al., 2023; Dubey et al., 2024; Shi et al., 2023a; Bolya et al., 2022; Jiang et al., 2022). Llama3 (Dubey et al., 2024) adopts the KV-Cache (Pope et al., 2023) to reduce the number of redundant calculations. AgentAttention (Han et al., 2023) employs the mediator token mechanisms to compress the interaction scale between Query and Key, yielding promising results in fundamental visual tasks.

ToMe (Bolya et al., 2022), TRIPS (Jiang et al., 2022), and CrossGET (Shi et al., 2023a) propose token merging strategies to reduce the number of tokens involved in global self-attention, thereby improving the efficiency of image, language, and multimodal understanding models. Similarly, DAM (Pu et al., 2024) and ToMeSD (Bolya & Hoffman, 2023) apply mediator token and token merging to Diffusion Transformer to save on computational costs, thereby reducing training overhead and inference time. However, token merging often leads to the loss of detailed information, which is particularly problematic for generative tasks that require strict detail preservation. Additionally, the compression of interaction scales between Query and Key in the mediator token mechanism can lead to a loss of important spatial relationships between neighboring tokens, which is crucial as illustrated in Sec. 3. To address these challenges, we propose a proxy token mechanism that establishes global associations through attention on a limited number of proxy tokens, guided by visual-spatial priors. At the same time, all tokens are retained to prevent the loss of detailed information, and the global associations in the proxy tokens are propagated to all detailed tokens. Furthermore, the modeling of spatially neighboring tokens is supported by window attention strategies.

## 3 METHOD

### 3.1 REDUNDANCY ANALYSIS

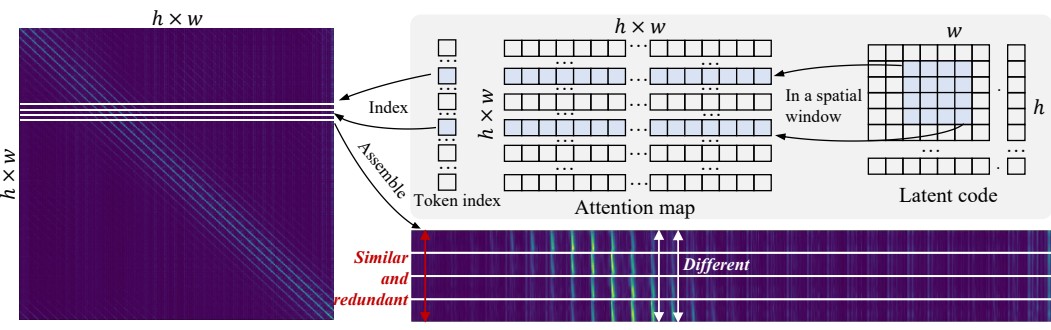

Figure 3: **The attention map of self-attention in PixArt-$\alpha$ at 512 resolution.** We assemble the attention map for 16 tokens within a $4 \times 4$ spatial window. The vertical axis represents different tokens within the window, and the horizontal axis represents their correlation with all latent tokens. It is evident that the attention of different tokens in the same window is almost identical for spatially distant tokens, whereas there is noticeable variation for spatially neighboring tokens.

Due to the sparsity and redundancy of visual information, global attention mechanisms in existing DiTs exhibit significant redundancy and computational complexity, particularly when processing high-resolution images and longer videos. We analyze this computational redundancy by visualizing the self-attention maps. Specifically, we examine the attention map of self-attention in PixArt-$\alpha$ at a resolution of $512 \times 512$, as shown on the left in Fig. 3. The attention map for latent codes within a spatial window is then assembled, as depicted on the right side of Fig. 3 (where the vertical axis represents different tokens in a window, and the horizontal axis represents the correlation with all latent tokens). It is evident that the attention maps for different tokens within the same window are nearly uniform for spatially distant tokens (i.e., at the same horizontal position, the vertical values are almost identical). Moreover, window tokens exhibit varying attention to spatially neighboring tokens. **This suggests that computing attention for all latent tokens is redundant, while attention for spatially neighboring tokens is critical.** Consequently, we propose a sparse attention strategy that samples limited proxy tokens from each window to perform self-attention, thereby reducing redundancy and decreasing complexity. Additionally, the association between spatially neighboring tokens is established through window attention. Further details are elaborated in Sec. 3.2.

### 3.2 ARCHITECTURE OF PT-DiT

As shown in Fig. 4, our proposed Proxy-Tokenized Diffusion Transformer (PT-DiT) introduces the proxy-tokenized mechanism to reduce the number of tokens involved in computing global self-attention, thereby efficiently establishing global visual associations. Specifically, the latent code $z \in \mathbb{R}^{C \times F \times H \times W}$ is passed through path embedding to obtain the latent code sequence $z_s \in \mathbb{R}^{N \times D}$. Subsequently, we add 3D positional encoding to $z_s$ and feed it into the well-designed

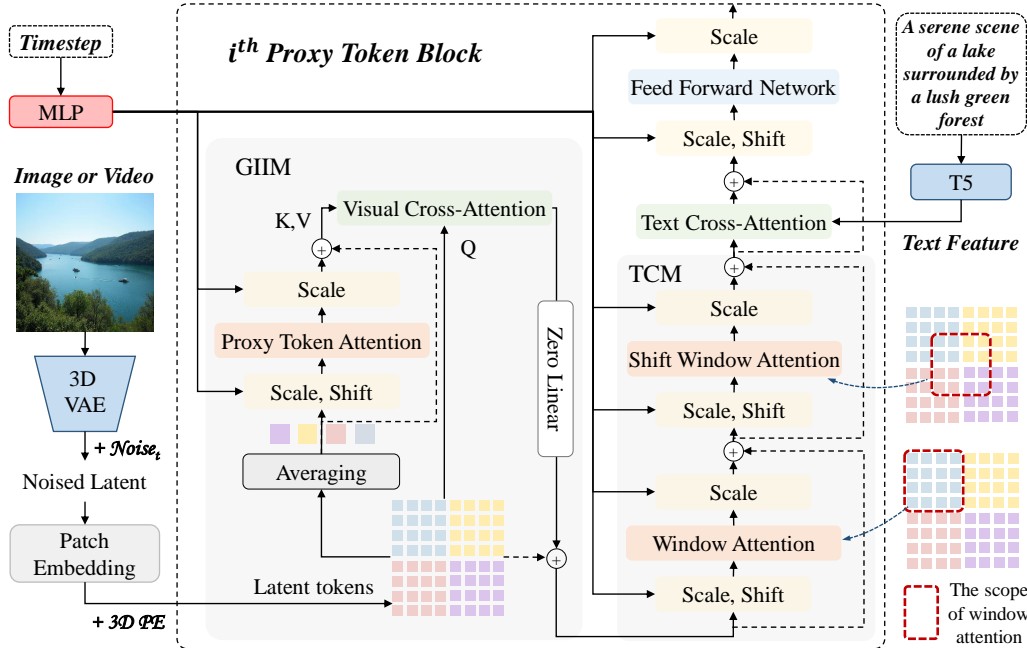

Figure 4: **The overall architecture of PT-DiT.** The image or video undergoes processing through a 3D VAE, followed by noise addition, patch embedding, and positional encoding to generate latent tokens. We replace global attention with proxy-tokenized attention to establish contextual associations and employ visual cross-attention to propagate this information to all tokens, thereby reducing computational redundancy. Moreover, texture detail modeling is enhanced through window attention and shifted window attention.

Proxy-Tokenized Blocks (PT-Block). Compared to the vanilla diffusion transformer block, the PT-Block introduces a Global Information Interaction Module (GIIM) and a Texture Complement Module (TCM). The GIIM facilitates efficient interaction among all latent codes using sparse proxy-tokenized mechanisms, while the TCM further refines local detail through window attention and shift-window attention. Below, we describe the GIIM and TCM in detail.

### 3.2.1 GLOBAL INFORMATION INTERACTION MODULE

Given a series of latent tokens, we first sample a set of proxy tokens based on their spatial and temporal priors. Each proxy token represents a localization within the image or video and interacts with proxy tokens in other regions to establish global relationships. Then, the information contained in proxy tokens is propagated to latent tokens, enabling efficient global visual information interaction.

Specifically, we reshape the latent code sequence $z_s \in \mathbb{R}^{N \times D}$ to $z_s \in \mathbb{R}^{f \times h \times w \times D}$, where $f$, $h$ and $w$ denotes the frame, height, and width of video or image ($f = 1$) in the latent space after patch embedding, thereby recovering its temporal and spatial connections. The set of proxy tokens $P_a \in \mathbb{R}^{D \times \frac{f}{p_t} \times \frac{h}{p_h} \times \frac{w}{p_w}}$ is calculated from each window of size $p_t \times p_h \times p_w$ using the averaging operation. The parameters $p_f$, $p_h$, and $p_w$ indicate the compression ratios for frame, height, and width, respectively. Each proxy token represents $p_t \times p_h \times p_w$ tokens, modeling global information with the other proxy token through self-attention. Subsequently, cross-attention is performed to propagate the global visual information into all latent tokens $z_s$, where the $z_s$ serves as the Query and the proxy tokens $P_a$ serve as the Key and Value. The above process is mathematically expressed as follows:

$$z_s = \mathrm{CS}(z_s, \mathrm{SA}(\mathrm{Averaging}(z_s)), \tag{1}$$

where $\mathrm{Averaging}(\cdot)$ refers to the averaging operation applied to tokens within the same window to extract proxy tokens, and $\mathrm{CS}(\cdot, \cdot)$ and $\mathrm{SA}(\cdot)$ represent the cross-attention and self-attention operations, respectively. Besides, we introduce a linear layer with zero initialization to enhance training stability. This approach allows the PT-Block to achieve efficient global information modeling and

avoids the computational overhead caused by redundant computations in self-attention. We will analyze the computational complexity advantages of GIIM further in Sec. 3.3.

### 3.2.2 TEXTURE COMPLEMENT MODULE

Due to the characteristics of the sparse proxy tokens interactions, the model's capacity to capture detailed textures is limited, making it challenging to meet the high-quality demands of generation tasks. To solve this problem, we introduce localized window attention, as proposed in Liu et al. (2021), which models texture information by computing attention only within a local spatial window of the image. This mechanism is incorporated into the Texture Complement Module (TCM) due to its effectiveness in detail modeling and computational efficiency. Specifically, the latent tokens $z_s$ are reshaped to $z_s \in \mathbb{R}^{\frac{f \times h \times w}{p_t \times p_h \times p_w} \times (p_t \times p_h \times p_w) \times D}$, where $\frac{f \times h \times w}{p_t \times p_h \times p_w}$ denotes the number of window in a image. Window self-attention is computed along the second dimension. To further enhance the model, shift-window attention is integrated into TCM, which applies spatial translation to the window divisions, enabling connections between neighboring tokens across different windows and mitigating the "grid" phenomenon caused by localized window attention. The formula for this process is as follows:

$$\hat{z_s} = \text{WSA}(z_s) + z_s,$$
$$z_w = \text{SWSA}(\hat{z_s}) + \hat{z_s}, \tag{2}$$

where $\text{SWSA}(\cdot)$ and $\text{WSA}(\cdot)$ denote shift-window attention and window attention respectively. Both window attention and shift-window attention introduce a visual prior to DiT, which aids in the construction of texture details and advances the training of visual generators. Moreover, the increase in computation is minimal due to the limited number of tokens in each window. We will analyze this in detail in Sec. 3.3. Then, $z_w$ is reshaped to $z_w \in \mathbb{R}^{N \times D}$ and fed into Textual Cross-Attention and MLP, similar to DiT.

### 3.2.3 COMPRESSION RATIOS

**For the image generation task**, we first determine the compression ratio at a resolution of 256. At this resolution, after applying the VAE ($8\times$ down-sampling) and patch embedding ($2\times$ down-sampling), the image is reduced to only $16 \times 16$ tokens. With a compression ratio of $(1, 4, 4)$, the number of windows in the space becomes $4 \times 4$, and the number of proxy tokens is 16. With this configuration, the limited number of windows and the large coverage area of each window make it challenging for a single proxy token to effectively represent the complex information within the window for global information modeling. This leads to anomalies in the image layout and performance degradation, as shown in the Table. 2(d). Setting the compression ratio to $(1, 2, 2)$ maintains a reasonable number of windows while preserving the necessary semantic richness within each window. Meanwhile, maintaining the same number of windows across different resolutions benefits the model's training process from low-to-high resolutions, as it ensures a consistent semantic hierarchy across resolutions. Therefore, the compression ratios $(p_f, p_h, p_w)$ is set to $(1, 2, 2)$, $(1, 4, 4)$, $(1, 8, 8)$, and $(1, 16, 16)$ at 256, 512, 1024, and 2048 resolution respectively. It is worth noting that when the input is an image, $f$ and $p_f$ will be set to 1.

**For the video generation task**, we set $p_f = 4$ across different resolution to maintain the temporal compression consistent. Owing to token compression in the frame, height and width dimensions, PT-DiT can effectively train a generator for longer videos.

### 3.3 COMPLEXITY ANALYSIS

With a small number of representative token attention, PT-DiT reduces the computational redundancy of the original full token self-attention. The advantages of our method in terms of computational complexity are further analyzed theoretically in the following.

The computational complexity of self-attention is $2N^2D$, computed as follows:

$$z = \text{Softmax}(z^{(q)}z^{(k)\top}/\sqrt{D})z^{(v)},$$
$$complexity = \underbrace{N^2D}_{z^{(q)}z^{(k)\top}:\mathbb{R}^{N \times D}\cdot\mathbb{R}^{D \times N}} + \underbrace{N^2D}_{Softmax(\cdot)z^{(v)}:\mathbb{R}^{N \times N}\cdot\mathbb{R}^{N \times D}} + \mathcal{O}(N^2), \tag{3}$$

where $N$ denotes the length of latent tokens and $D$ represents feature dimension. Similarly, the computational complexity of GIIM and TCM is computed as follow:

$$complexity = 2\underbrace{\frac{N^2}{(p_f p_h p_w)^2}D}_{\text{SA in GIIM}} + 2\underbrace{\frac{N^2}{p_f p_h p_w}D}_{\text{CS in GIIM}} + 4\underbrace{\frac{N}{p_f p_h p_w}(p_f p_h p_w)^2 D}_{\text{WSA and SWSA in TCM}}$$
$$= 2\left(\frac{1}{(p_f p_h p_w)^2} + \frac{1}{p_f p_h p_w} + \frac{2p_f p_h p_w}{N}\right)N^2 D. \tag{4}$$

Obviously, due to the proxy-tokenized strategy, our method provides significant advantages, especially with larger compression ratios ($p_f$, $p_h$, $p_w$) and longer sequence lengths ($N$). When ($p_f$, $p_h$, $p_w$) are (1, 2, 2), (1, 4, 4), (1, 8, 8), and (1, 16, 16) and the image resolution are 256 ($N = 256$), 512 ($N = 1024$), 1024 ($N = 4096$), and 2048 ($N = 16348$), our method accounts for only 34.3%, 9.7%, 4.7%, and 2.3% of the total self-attention. In addition, PT-DiT offers even greater benefits for video generation tasks with longer sequence lengths. Experimental analysis is available in Sec. 4.4.

## 4 EXPERIMENT

### 4.1 EXPERIMENTAL SETUP

**Training Setting.** Due to limitations in computational resources, we only trained PT-T2I and PT-T2V based on PT-DiT/XL 1.1B. Following previous methods (Xu et al., 2024; Yang et al., 2024; Chen et al., 2023), we utilize the T5 large language model as the text encoder and train PT-T2I using a low- to high-resolution strategy divided into three stages. Detailed hyper-parameter settings and the model configurations for various PT-DiT scales are provided in **Appendix.** A.2.

**Ablation Study.** We conduct ablation experiments using a class-conditional version of PT-DiT/S-Class (32M) on the ImageNet (Deng et al., 2009) benchmark at 256 resolution. The AdamW optimizer is utilized with a constant learning rate of 1e-4. We train the models for 400,000 iterations with a batch size of 256, while maintaining an exponential moving average (EMA) of the model weights. During inference, we set the denoising step as 50 and use classifier-free guidance (cfg=6.0).

### 4.2 QUALITATIVE ANALYSIS

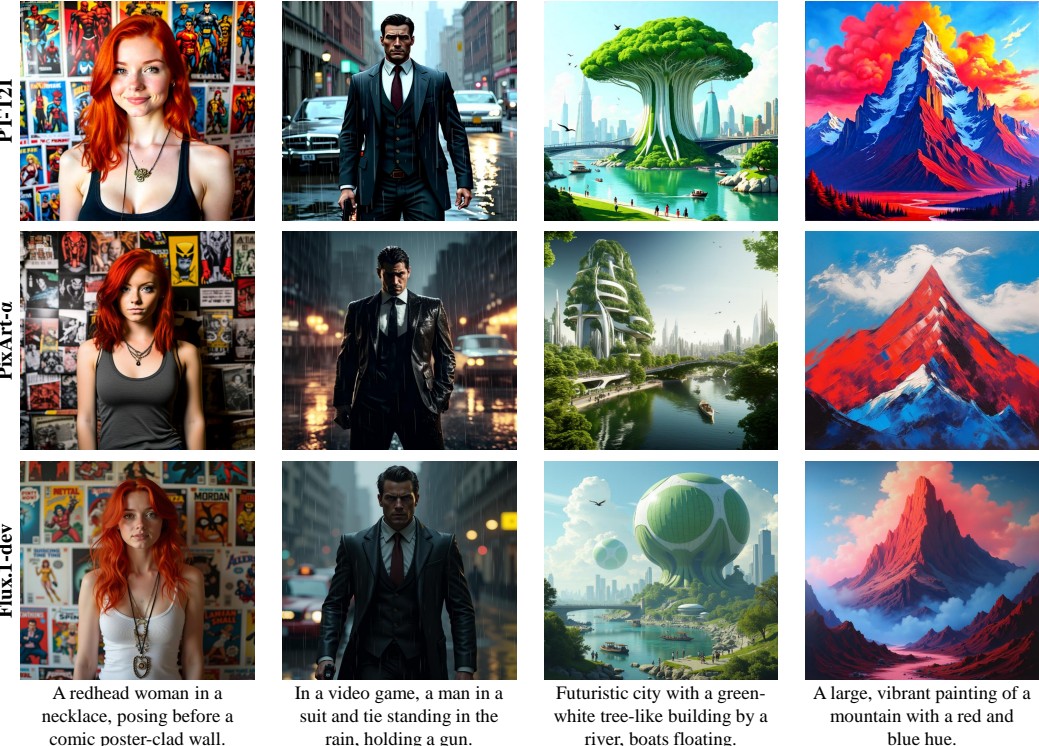

Figure 5: Qualitative comparison of Text-to-Image generation models.

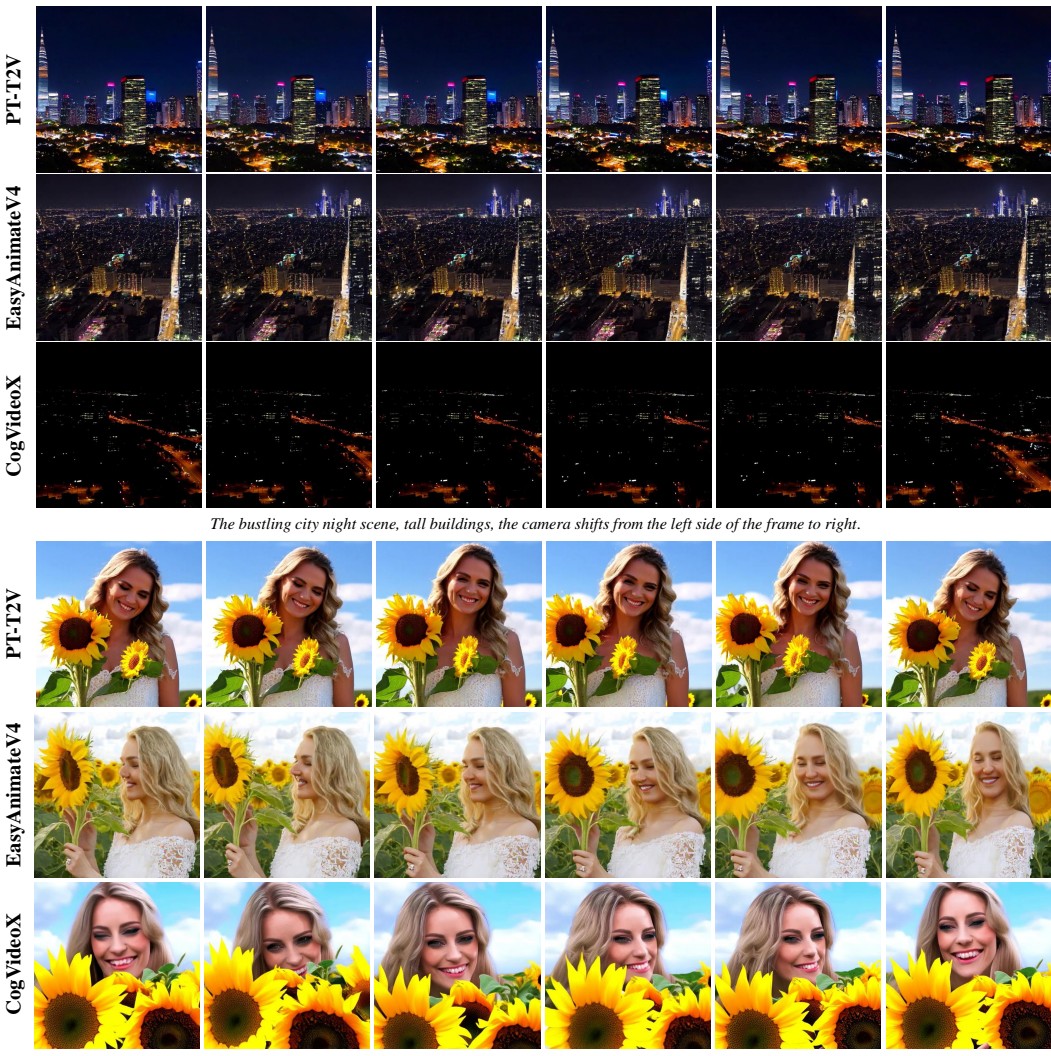

*The bustling city night scene, tall buildings, the camera shifts from the left side of the frame to right.*

*Blonde woman with sunflowers, smiling in a sunflower field under blue sky.*

Figure 6: Qualitative comparison of Text-to-Video generation models.

**Text-to-Image.** We provide a qualitative comparison of PT-T2I with existing state-of-the-art Text-to-Image models (e.g., PixArt-$\alpha$ and Flux) at a resolution of 1024, as shown in Fig. 5. PT-T2I exhibits competitive performance, generating photo-realistic images that align well with the provided text prompts. Additional samples generated by PT-T2I can be found in Anonymous Repository.

**Text-to-Video.** We also compare PT-T2V with the recently released open-source Text-to-Video models (i.e., EasyAnimateV4 and CogVideoX) at a resolution of 512, achieving comparable results, as depicted in Fig. 6. More video samples are available in the Anonymous Repository.

**Text-to-MV.** Please refer to **Appendix.** A.5

### 4.3 QUANTITATIVE ANALYSIS

**MS-COCO.** We conduct experiments to quantitatively evaluate PT-T2I using zero-shot FID-30K on the MS-COCO (Lin et al., 2014) $256 \times 256$ validation dataset, as shown in Table 1(a). Due to the distribution gap between our collected data and MS-COCO, there is a resulting decrease in FID (Heusel et al., 2017) metrics. Nevertheless, PT-T2I achieves a competitive score of 15.70.

**MSR-VTT and UCF-101.** We evaluate PT-T2V on two standard video generation benchmarks, MSR-VTT (Xu et al., 2016) and UCF-101 (Soomro et al., 2012), at a resolution of 256. As shown in Table 1(b), PT-T2V achieves state-of-the-art results among DiT-based approaches and demonstrates competitive performance compared to U-Net-based approaches. Notably, since CogVideoX,

EasyAnimateV4, and PT-T2V all utilize T5 as the text encoder, this creates a gap in CLIPSIM compared to methods that employ the CLIP as the text encoder, such as AnimateDiff, DynamiCrafter, PixelDance, and FancyVideo.

Table 1: The quantitative evaluation of the Text-to-Image (a) and Text-to-Video (b) tasks.

(a) Quantitative evaluation on the MS-COCO FID-30K scores (zero-shot).

| Method | FID-30k↓ |
|---|---|
| DALL-E 2 (Ramesh et al., 2022) | 10.39 |
| SD (Rombach et al., 2022) | 8.73 |
| Imagen (Saharia et al., 2022) | 7.27 |
| RAPHAEL (Xue et al., 2024) | 6.61 |
| Kolors (Team, 2024) | 23.15 |
| PixArt-$\alpha$ (Chen et al., 2023) | 10.65 |
| Flux.1-dev(BlackForestlabs AI, 2024) | 22.76 |
| PT-T2I | 15.70 |

(b) Quantitative evaluation on the UCF-101 (Soomro et al., 2012) and MSR-VTT (Xu et al., 2016). The best and second performing metrics are highlighted in **bold** and underline respectively.

| Method | Arc | Data | UCF-101 | | | MSR-VTT | |
|---|---|---|---|---|---|---|---|
| | | | FVD(↓) | IS(↑) | FID(↓) | FVD(↓) | CLIPSIM(↑) |
| AnimateDiff (Guo et al., 2023) | U-Net | 10M | 584.85 | 37.01 | 61.24 | 628.57 | 0.2881 |
| DynamiCrafter (Xing et al., 2023) | U-Net | 10M | 404.50 | 41.97 | **32.35** | 219.31 | 0.2659 |
| PixelDance (Zeng et al., 2024) | U-Net | 10M | 242.82 | 42.10 | 49.36 | 381.00 | **0.3125** |
| FancyVideo (Feng et al., 2024) | U-Net | 10M | 412.64 | **43.66** | 47.01 | 333.52 | 0.3076 |
| CogVideoX-2B(Yang et al., 2024) | DiT | 35M | 680.11 | 33.44 | 62.57 | 418.14 | 0.2318 |
| EasyAnimateV4 (Xu et al., 2024) | DiT | 12M | 694.80 | **44.09** | 92.33 | 568.99 | 0.2285 |
| PT-T2V | DiT | 10M | **384.03** | 35.19 | 51.95 | **375.23** | 0.2349 |

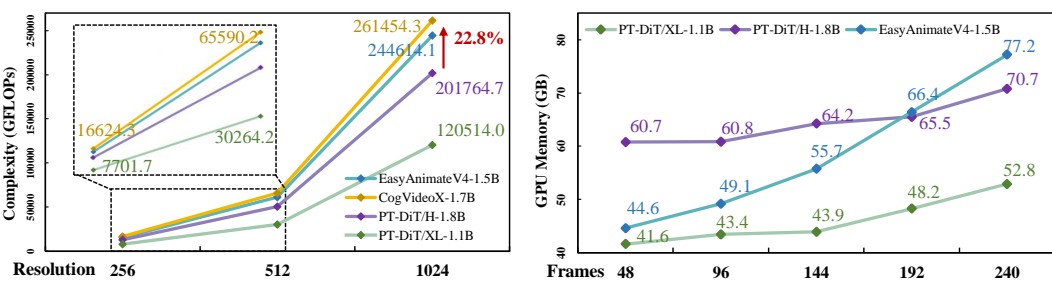

Figure 7: Comparison of video generation models in terms of GFLOPs and GPU memory usage.

## 4.4 ALGORITHMIC EFFICIENCY COMPARISON

As discussed in Sec. 3.3, our method effectively reduces complexity. In this section, we further analyze the computational advantages of PT-DiT in T2I and T2V tasks. For the T2I task, the results are reported in Fig.10 with related discussions referenced in **Appendix.** A.4.

In the video generation task, we assess our model based from two aspects: computational complexity and GPU memory consumption, as illustrated in Fig. 7. We conduct experiments using two scales of PT-DiT (i.e., PT-DiT/H (1.8B) for a consistent scale comparison and our utilized PT-DiT/XL (1.1B) for training PT-T2V) and select the latest open-source T2V model (i.e., CogVideoX-2B (actual test at 1.7B) and EasyAnimateV4 (1.5B)) as the comparison methods. The left side of Fig. 7 displays the GFLOPs calculated at different resolutions, with the latent code set to a time dimension of 48. It is obvious that, despite having the largest number of parameters, PT-DiT/H exhibits the lowest computational complexity. Meanwhile, the computational complexity of PT-DiT/XL employed by PT-T2V is only 50% that of CogVideoX and EasyAnimateV4. On the right side of Fig. 7, we further compare the GPU memory usage during training with EasyAnimateV4 at a resolution of 512, across different frame counts. Since the T2V version of EasyAnimateV4 employs HunyuanDiT with full 3D attention, its memory consumption increases dramatically with the number of video frames. In contrast, PT-DiT, which also utilizes 3D spatial-temporal modeling, experiences only a slight increase in memory consumption due to its well-designed proxy-tokenized attention mechanism. The above experiments demonstrate the potential of PT-DiT for generating longer and higher-resolution videos.

Table 2: Ablation study on PT-DiT/S-Class. Models are trained for 400k iterations.

(a) Major component.  (b) Proxy token extraction.  (c) Global information injection.  (d) Compressed ratio.

| Structure | FID-50k↓ | Method | FID-50k↓ | Method | FID-50k↓ | Ratio | FID-50k↓ |
|---|---|---|---|---|---|---|---|
| w/o GIIM | 23.71 | Average | 19.30 | Cross-Attention | 19.30 | 1, 2, 2 | 19.30 |
| w/o SWA | 23.59 | Top-Left | 20.84 | Interpolate | 21.82 | 1, 4, 4 | 21.24 |
| w/o TCM | 69.07 | Random | 21.00 | Linear | 20.24 | 1, 8, 8 | 20.43 |

## 4.5 ABLATION STUDY

**Major Component.** We conduct quantitative experiments to assess the effectiveness of the GIIM and TCM proposed in this paper. The absence of either GIIM or TCM results in a substantial performance loss (i.e., $19.30 \rightarrow 23.71$ or $19.30 \rightarrow 69.07$). Specifically, without TCM, the model struggles to capture fine details, making it challenging to meet generation tasks that demand high-quality detail, leading to a significant decline in performance. Additionally, we investigated the role of shift-window attention through both qualitative evaluation at a resolution of 512 and quantitative analyses at a resolution of 256, as illustrated in Fig. 8 and Table 2(a) respectively. As anticipated, there is a noticeable decrease (i.e., $19.30 \rightarrow 23.59$) in FID without shift-window attention, accompanied by pronounced "grid" phenomena.

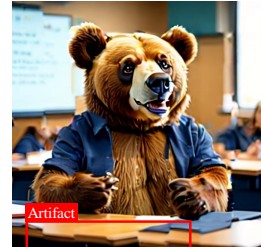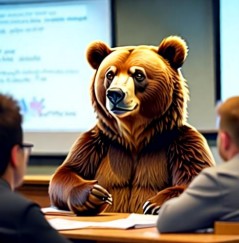

Without Shift-window Attention      With Shift-window Attention

Figure 8: Ablation on shift-window attention.

**Proxy Token Extraction.** As illustrated in Table 2(b), we explore three methods for obtaining the proxy token: the top-left token, a randomly selected token, and averaging the in-window tokens. A performance gap exists between the Top-Left (20.84) or Random (21.00) selections and the averaging manner (19.30). We believe this gap arises because the random and top-left tokens fail to adequately represent the overall characteristics of the region, compromising the effectiveness of proxy-tokenized attention and leading to performance loss. We use averaging as the default setting.

**Global Information Injection.** Due to the misalignment between the number of proxy tokens and latent tokens, we investigate three schemes for injecting global information into latent tokens: Cross-Attention, Interpolation, and Linear projection, as shown in Table 2(c). Among these, interpolation involves applying spatially bilinear interpolation to the proxy tokens, while linear projection aligns proxy tokens with latent tokens through a linear layer. Since each latent code can leverage global information from the entire set of proxy tokens, Cross-attention achieves a performance advantage with an FID of 19.30 compared to Linear projection at 20.24 and Interpolation at 21.82.

**Compressed Ratio.** As reported in Table 2(d), we examine the impact of compression ratio on performance at a resolution of 256. It is evident that when the compression ratio is high, the representative token fails to adequately capture the features of the region for effective global modeling, leading to a noticeable decline in performance (i.e., from 19.30 to 21.24 at (1, 4, 4)).

## 5 CONCLUSION

Given the sparsity and redundancy of visual information, this paper proposes PT-DiT, which leverages the proxy-tokenized attention mechanism to mitigate the computational redundancy of self-attention in diffusion transformers. A series of representative tokens are calculated based on temporal and spatial priors, with global interactions between them. Additionally, window attention and shifted window attention are introduced to refine the modeling of local details. Our proposed representative token mechanism is particularly effective for video tasks with redundant information, enabling 3D spatio-temporal modeling while avoiding an explosion in computational complexity. Experiments demonstrates that PT-DiT achieves competitive performance while delivering significant efficiency. We further develope the PT-T2X series based on PT-DiT, including models like T2I, T2V, and T2MV. We hope PT-DiT and PT-T2I/V can provide new insights and references for the field of diffusion transformers.

## ACKNOWLEDGEMENTS

This work was supported by National Key Research and Development Program of China (2024YFE0203100), National Natural Science Foundation of China (NSFC) under Grants No.62372482 and No.62476293, Nansha Key R&D Program under Grant No.2022ZD014, and General Embodied AI Center of Sun Yat-sen University.

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

# A APPENDIX

## A.1 RELATED WORK

**Image Generation with Diffusion Transformer.** Recent studies (Peebles & Xie, 2023; Ma et al., 2024a; Li et al., 2024b;a) have demonstrated the potential of using the Vision Transformer (ViT) (Han et al., 2022) as an alternative backbone for image generation. U-ViT (Bao et al., 2023) encodes the condition as tokens and incorporates skip connections inspired by U-Net, achieving excellent performance. DiT (Peebles & Xie, 2023) introduces AdaLN to integrate conditions and analyzes the scalability, complexity, and performance of ViT in comparison to U-Net. SiT (Ma et al., 2024a) adds an interpolant framework to DiT, achieving even better scores on ImageNet (Deng et al., 2009). PixArt-$\alpha$ (Chen et al., 2023) integrates cross-attention modules into DiT to inject text conditions and optimize the class-conditional branch. Flag-DiT (Gao et al., 2024) and Next-DiT (Zhuo et al., 2024) employ advanced techniques like RoPE (Su et al., 2024), RMSNorm (Zhang & Sennrich, 2019), and flow matching (Lipman et al., 2022) to enhance stability, and they use zero-initialized attention to incorporate complex text instructions. Although the effectiveness of transformers in diffusion models has been validated, the substantial computational and spatial complexities of these models still need to be addressed. DAM (Pu et al., 2024) has drawn attention to the redundancy within DiT and uses the mediator tokens to directly proxy the query and key in the attention operation, breaking down the process into two distinct attention calculations. We propose PT-DiT, a method that employs a more gentle strategy for compressing hidden states. This approach not only reduces redundancy but also guarantees a minimal loss of information during the compression process, thereby substantially decreasing the computational complexity.

## A.2 TRAINING DETAIL AND MODEL CONFIGURATION

We collect a total of 50M data points for the training set, including 32M images with an aesthetic score of 5.5 or higher from Laion (Schuhmann et al., 2022) and 18M high-resolution, high-quality datasets that we constructed. During the high-resolution training phase, we exclusively use 18M high-quality data. We train PT-T2V by progressing through three stages starting from stage 1 of PT-T2I, with detailed hyper-parameters shown in Table 3. The WebVid 10M (Bain et al., 2021) dataset is employed as the 256-resolution video training data. Additionally, we collect 3M high-resolution, high-quality video samples from the Internet to train the high-resolution video generator. The training objective for PT-T2I/V is v-prediction, with an extracted text token length of 120. During the inference phase, the denoising steps are set to 50, and the scale of classifier-free guidance is set to 6.0. The specific parameter configurations for various scales of PT-DiT are presented in Table 4.

Table 3: The training setups of PT-T2I and PT-T2V

| Text-to-Image | | | | | Text-to-Video | | | | |
|---|---|---|---|---|---|---|---|---|---|
| Resolution | Data | Learning Rate | Batch Size | Iteration | Resolution # Frame | Data | Learning Rate | Batch Size | Iteration |
| 256 | 50M | 2e-5 | 10240 | 100k | - | - | - | - | - |
| 512 | 18M HQ | 2e-5 | 768 | 50k | 256 # 96 | 10M | 2e-5 | 512 | 100k |
| 1024 | 18M HQ | 2e-5 | 512 | 50k | 512 # 96 | 3M HQ | 2e-5 | 256 | 50k |

Table 4: The model configurations for various scales of PT-DiT.

| Model | Layers | Hidden Dim | Head Number | Param. (M) |
|---|---|---|---|---|
| PT-DiT/S-Class | 10 | 288 | 6 | 32 |
| PT-DiT/B-Class | 12 | 576 | 8 | 154 |
| PT-DiT/B | 12 | 640 | 10 | 144 |
| PT-DiT/L | 28 | 864 | 12 | 605 |
| PT-DiT/XL | 28 | 1152 | 16 | 1142 |
| PT-DiT/H | 30 | 1440 | 20 | 1795 |

## A.3 PERFORMANCE AND EFFICIENCY COMPARISON

To fairly compare the performance of PT-DiT with DAM and DiT, we conduct the widely used class-to-image experiments on ImageNet 256. Since DAM is trained on the SiT codebase, we utilize PT-DiT/S-Class to align with DiT-S, SiT-S, and DAM-S, using two different codebases (DiT-based and SiT-based) with the same parameters, training data, training steps (400k), and experimental configurations. The detailed model settings are provided in Table. 4, and the results are shown in the Fig. 9, with the data reported without the use of Classifier-free guidance (CFG).

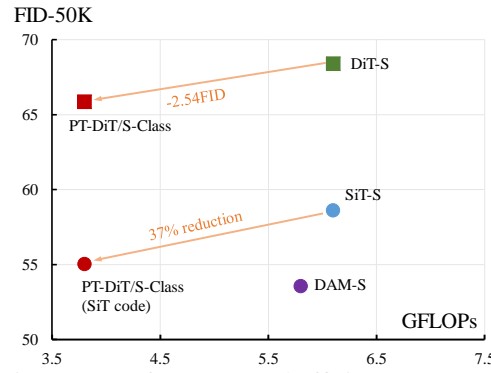

Figure 9: Performance and efficiency comparison on ImageNet at a resolution of 256.

Compared to DiT and SiT, PT-DiT achieves competitive performance and a significant efficiency advantage, thanks to the well-designed proxy token mechanism and texture refinement module. For instance, PT-DiT/S-Class achieves the FID of 65.86 ($\downarrow$ 2.54 FID) and 55.05 ($\downarrow$ 3.56 FID), outperforming DiT-S (68.40 FID) and SiT (68.61 FID). Meanwhile, the computational complexity is reduced by 37% (3.8 vs. 6.1 GFLOPs). These results demonstrate that our method can maintain competitive performance while benefiting from reduced computational complexity. This advantage is primarily due to the window-attention, which aligns with visual priors and models the spatial neighboring token, as well as the efficiency gains from the proxy token mechanism. Furthermore, when compared to DAM, PT-DiT achieves similar performance while offering a significant reduction in computational complexity (i.e., 3.8 GFLOPs for PT-DiT vs. 5.8 GFLOPs for DAM).

## A.4 ALGORITHMIC EFFICIENCY COMPARISON

In the image generation task, similar to Fig. 2, we conduct comparisons at different parameter scales. With equivalent parameter counts, we compared Lumina-Next (1.7B) to our PT-DiT/H (1.8B), DiT/B (0.13B) and DAM/B (0.13B) to our PT-DiT/B-Class (0.14B), as illustrated in Fig. 10. As shown on the left side of Fig. 10, the GFLOPs of PT-DiT/H are significantly lower than Lumina-Next across multiple scales. Specifically, at resolutions of 512 and 2048, PT-DiT/H achieves complexity reduction of respectively 82.0% and 82.5%. Similarly, the right side of Fig. 10 indicates that PT-DiT/B-Class requires 59.5% less computation than DiT/B at a resolution of 1024. Compared to DAM/B, which has an attention computation complexity of $O(n)$, our method exhibits a comparable level of computation complexity across all resolutions.

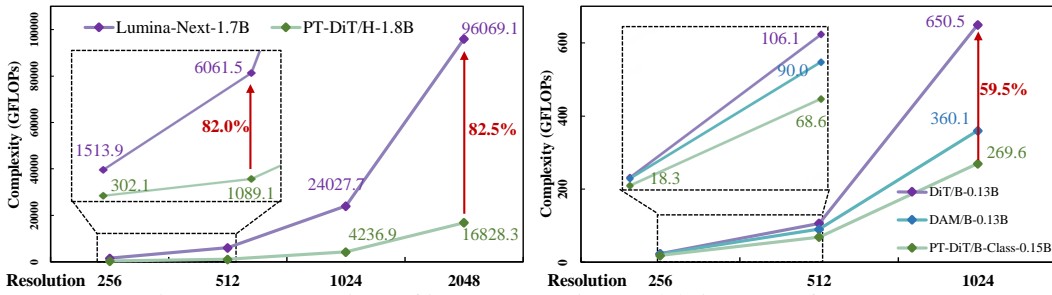

Figure 10: Comparison of image generation models in terms of GFLOPs.

## A.5 PT-T2MV

**Text-to-MV.** We further explore the effectiveness of PT-DiT on Text-to-MultiView (T2MV) tasks. The trained PT-T2MV is capable of generating $512 \times 512 \times 24$ images from various viewpoints based on the provided text instruction, showcasing strong spatial consistency, as illustrated in Fig. 11. The detailed experimental and training setups are as follow.

**Basic setting.** MultiView images of 3D objects can be interpreted as videos of static objects. We utilize a subset of approximately 40k samples from G-Objaverse (Qiu et al., 2024), following VideoMV

(Zuo et al., 2024), which is rendered as video data to train our PT-T2MV model. Each object is rendered with a uniformly distributed azimuth from 0° to 360° and an elevation ranging from 5° to 30°, resulting in a $512 \times 512 \times 24$ video.

**Training setting.** Following previous works (Zuo et al., 2024; Shi et al., 2023b), we only accept text instruction as input to generate the Multi-View images of 3D object without additional reference images and camera parameters. The PT-T2MV is trained from stage 2 of the PT-T2I, with a bacthsize of 128 and 20k iterations. The other hyperparameters and experimental settings are the same as in PT-T2I.

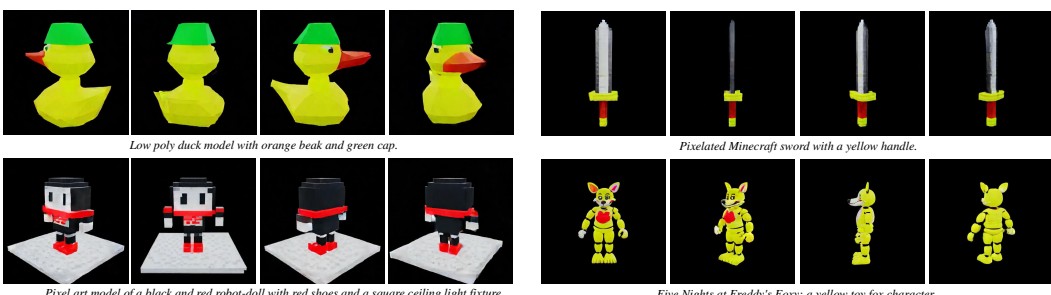

*Low poly duck model with orange beak and green cap.*

*Pixelated Minecraft sword with a yellow handle.*

*Pixel art model of a black and red robot-doll with red shoes and a square ceiling light fixture.*

*Five Nights at Freddy's Foxy: a yellow toy fox character.*

Figure 11: Samples by PT-T2MV. It is important to note that PT-T2MV does not accept any image inputs or camera parameters and relies solely on text prompts.

