# OpenReview forum: "PT-T2I/V: An Efficient Proxy-Tokenized Diffusion Transformer for Text-to-Image/Video-Task"
_ICLR.cc/2025/Conference — ICLR 2025 Poster_

### Official Review · Reviewer_q44Z · 2024-10-29

**Soundness:** 3
**Presentation:** 3
**Contribution:** 4
**Rating:** 8
**Confidence:** 5

**Summary:**

This paper first highlights the computational redundancy issue of global self-attention in diffusion transformers, stemming from sparse visual information, through an analysis of the attention map. To tackle this problem, the authors propose the Proxy-tokenized Diffusion Transformer (PT-DiT), which substitutes the global self-attention of the vanilla DiT with Global Information Interaction Module (GIIM) and Texture Complement Module (TCM), enabling efficient modeling of both global and local visual information. In experiments, PT-DiT demonstrates theoretical and experimental advantages in computational complexity and shows competitive performance across various tasks, including T2I, T2V, and T2MV, underscoring its generalization capabilities.

**Strengths:**

-The motivation for reducing the training and inference costs of DiT by minimizing redundant computation in global self-attention is clear.

-The GIIM and TCM design in PT-DiT is promising and straightforward to implement.

-Qualitative and quantitative experiments on a variety of tasks (e.g., Text-to-Image, Text-to-Video, and Text-to-MultiView) are comprehensive and convincing.

**Weaknesses:**

-This article lacks results on ImageNet for a fair comparison with existing state-of-the-art diffusion transformer models. Previous studies, such as DiT, MDT, VAR, and SiT, have all been evaluated on ImageNet using metrics like FID and IS. To fully establish its effectiveness, PT-DiT should also be assessed under similar conditions. This is essential for demonstrating its competitive performance despite reduced computational complexity and should be included in evaluations.

-How is the hyper-parameter compression ratio determined in this paper? Why is the compressed ratios set to (1, 2, 2) at a resolution of 256, and will a ratio of (1, 16, 16) result in excessive information loss at a resolution of 1024? The authors should provide further clarification on the rationale and motivations behind the chosen compressed ratios settings in their methodology.

-The extraction of proxy tokens through averaging is straightforward, but how representative are these proxy tokens when obtained in this manner? Could proxy tokens also be extracted using clustering or similarity-based methods for potentially improved representativeness?

**Questions:**

-What accounts for PT-DiT's 80% improvement over Lumina in Figure 8 (left), especially since this enhancement is significantly greater than the improvements observed for other models?

-How about the training time and inference time of PT-DiT?

-Does the author have any plans to open source the Qihoo-T2X series of models and code?

---

> ### Author Response · Authors · 2024-11-22
> **Response to Reviewer q44Z (Part 1)**
>
> **General Reply**
>
> Thank you very much for your valuable comments. Below, we provide detailed responses to each of your questions and comments. If you have any further questions or concerns regarding our paper or our responses, please let us know, and we will address them promptly.
> ___
> **W1: A Fair Comparison on ImageNet**
> > This article lacks results on ImageNet for a fair comparison with existing state-of-the-art diffusion transformer models. Previous studies, such as DiT, MDT, VAR, and SiT, have all been evaluated on ImageNet using metrics like FID and IS. To fully establish its effectiveness, PT-DiT should also be assessed under similar conditions. This is essential for demonstrating its competitive performance despite reduced computational complexity and should be included in evaluations.
>
> **A1:** Thanks for your valuable suggestion. Following your advice, we conduct experiments on ImageNet at a resolution of 256, aligning PT-DiT/L-Class (693M) with DiT-XL (675M), and ensuring consistent training and inference setups for a fair comparison. Due to time constraints, we train for 2,000,000 steps. For evaluation, we randomly sample 50,000 images for testing and assessed performance using metrics such as FID and IS. The results, shown below, demonstrate that, thanks to the well-designed agent token mechanism and TCM, our method achieves competitive performance with DiT-XL while offering a clear efficiency advantage.
>
> | Model | Step | Params(M) | GFLOPs | FID | sFID | IS | Precision | Recall |
> | ----------- | ----------- | ----------- |----------- |----------- |----------- |----------- |----------- |----------- |
> | DiT-XL | 7000k | 675 | 118.7 | 9.62 | 6.85 | 121.50 | 0.67 | 0.67 |
> | SiT-XL | 7000k | 675 | 118.7 | 8.3| - | - | - | - |
> | PT-DiT/L-Class   | 2000k | 693 | 82.6 | 9.40 | 6.17 | 125.07 | 0.68 | 0.66 |
> ___
> **W2: The Value of  Compression Ratio**
> > How is the hyper-parameter compression ratio determined in this paper? Why is the compressed ratios set to (1, 2, 2) at a resolution of 256, and will a ratio of (1, 16, 16) result in excessive information loss at a resolution of 1024? The authors should provide further clarification on the rationale and motivations behind the chosen compressed ratios settings in their methodology.
>
> **A2:** Thank you for your insightful suggestion.
>
> **We first determine the compression ratio at a resolution of 256**. At this resolution, after applying the VAE (8x down-sampling) and patch embedding (2x down-sampling), the image is reduced to only 16x16 tokens. With a compression ratio of (1, 4, 4), the number of windows in the space becomes 4x4, and the number of proxy tokens is 16. We found that, with this configuration, the limited number of windows and the large coverage area of each window make it challenging for a single proxy token to effectively represent the complex information within the window for global information modeling. This leads to anomalies in the image layout and performance degradation, as shown in the Table. 2(d) in the paper. Setting the compression ratio to (1, 2, 2) maintains a reasonable number of windows while preserving the necessary semantic richness within each window. Through experimentation, we found that this setting strikes a balance between maintaining performance and improving efficiency. In contrast, the (1, 1, 1) setting does not support proxy token compression or window-attention, which are crucial for reducing computational complexity. As a result, we chose a compression ratio of (1, 2, 2) for the 256 resolution.
>
> **For higher resolutions**, we found that maintaining the same number of windows across different resolutions benefits the model's training process, as it ensures a consistent semantic hierarchy across resolutions.
>
> By continuously preserving the detail tokens, a higher compression fraction does not result in a loss of information. Following your suggestion, we will revise this section in the paper for clarity and highlight the changes in red.

---

> ### Author Response · Authors · 2024-11-22
> **Response to Reviewer q44Z (Part 2)**
>
> **W3: The Extraction of Proxy Tokens**
> > The extraction of proxy tokens through averaging is straightforward, but how representative are these proxy tokens when obtained in this manner? Could proxy tokens also be extracted using clustering or similarity-based methods for potentially improved representativeness?
>
> **A3:** Thank you for your suggestion. Extracting proxy tokens based on similarity may introduce additional computation, which could reduce the efficiency of PT-DiT. We conduct ablation experiments to explore the impact of similarity-based proxy token selection. Specifically, during each sampling, the cosine similarity of each token with all other tokens in the window is calculated after L2 normalization, and the token with the highest sum of similarities is chosen as the proxy token. The results, shown below, indicate that while the similarity-based approach provides only a modest performance improvement, it also introduces greater computational complexity. To strike a better balance between computational efficiency and performance, the average-based approach remains the more suitable option.
>
> | Method | FID-50K | Params (M) | GFLOPs|
> | ----------- | ----------- | ----------- |----------- |
> | Average  | 19.30 | 32.6 | 3.82 |
> | Similarity | 19.25 | 32.6 | 4.01 |
> ___
> **Q1:** What accounts for PT-DiT's 80% improvement over Lumina in Figure 8 (left), especially since this enhancement is significantly greater than the improvements observed for other models?
>
> **A4:** Thank you for your question. We believe this difference is due to the specific design of the Lumina-Next structure. With a higher number of transformer blocks (1152, compared to just over 30 in PT-DiT/H), Lumina-Next includes more attention operations. As a result, PT-DiT, which is based on the proxy token mechanism, can offer more significant advantages in terms of efficiency.
> ___
> **Q2:** How about the training time and inference time of PT-DiT?
>
> **A5:** Thank you for your question. We test the training and inference times of PT-DiT/S-Class, and as expected, our method successfully accelerates both model training and inference, as demonstrated by the following results.
>
> | Method | Training time (steps/s, with 32 batch size) | Inference time (steps/s, with 1 batch size) |
> | ----------- | ----------- | ----------- |
> | DiT-S  | 3.6 | 11.76 |
> | PT-DiT/S-Class | 4.3 | 16.12 |
> ___
> **Q3:** Does the author have any plans to open source the Qihoo-T2X series of models and code?
>
> **A6:** Thank you for your interest. We are open-sourcing the entire Qihoo-T2X family of models and code to support the development of the efficient diffusion transformer.

---

> > ### Comment · Reviewer_q44Z · 2024-11-24
> > **Maintaining the rate**
> >
> > Thank you for your thoughtful response. Your answers have resolved all my questions. I hope the relevant discussions can be included in the final version of the paper.

---

### Official Review · Reviewer_wrGZ · 2024-11-03

**Soundness:** 3
**Presentation:** 3
**Contribution:** 3
**Rating:** 6
**Confidence:** 5

**Summary:**

This paper introduces the Qihoo-T2X series, featuring the Proxy-Tokenized Diffusion Transformer (PT-DiT) for efficient text-to-video generation. PT-DiT uses proxy-tokenized attention to reduce computational redundancy by selecting representative tokens within spatial-temporal windows, enhancing global information capture without the high complexity of full self-attention. This design integrates global semantics efficiently through sparse proxy tokens and refines texture details using window and shifted-window attention, similar to the Swin Transformer. The Qihoo-T2X family demonstrates competitive performance with reduced complexity compared to models like DiT and PixArt-α.

**Strengths:**

- The paper is well-written and easy to follow.
- The visualizations are promising.
- Compared with SOTA methods, the model can achieve a similar level of performance with less training data. Meanwhile, the inference cost is much lower.

**Weaknesses:**

- The method section is not clear enough. The mechanisms of window attention and shifting window attention are still not clear.
- The task of T2MV is a little strange. What would be an application scenario for that task?
- The "text-to-any-task" is overclaimed. The focused three tasks are essentially video generation (with a single frame or multi-view video).

See the next question sections for more details.

**Questions:**

About method section:
-  l201-203, f, h, w are not the actual frame, height, and width of the video/image, but the latent, right? Also, for images, how does the 3D VAE, with temporal downampling layers, handle them?
- the texts from l242 to l244 are confusing, $(p_f \cdot p_f, p_h)$ should be ($p_f, p_h, p_w$)? so we have $p_h = p_w$ = <some value depends on resolution> and $p_f$ = 1 for image and $p_f$ = 4 for videos, right?
- Details for the window attention and shifting window attention are missing. Only Figure 4 shows some fundamental ideas of them but how do they work exactly? How does the shifting window work?

About T2MV:
- Why Text-to-MV task is interesting? At the training level, there is no difference from this task to Text-to-video unless the training data; while for real practice, it's hard to control the generated content from only texts without any reference image.
- It's hard to evaluate the performance of T2V-MV. Can the T2V-MV work for out-of-distribution prompts? And how about the visual and quantitative evaluations with other models, given that the task setting is following previous approaches?

Other questions:
- The page limit is 10, no need to move related work to the appendix.
- Can we use a VAE with a higher compression ratio, or a large patch embedding to achieve a similar level of efficiency? In principle, with the proposed design, more details could be restored compared with directly using a higher compression ratio (but with vanilla attention). It would also be great to show that.

**Details Of Ethics Concerns:**

The model name "Qihoo-T2X" has nothing to do with the proposed PT-DIT but potentially indicates the identity of the researchers. The authors also did not explain the model name in the paper.

---

> ### Author Response · Authors · 2024-11-22
> **Response to Reviewer wrGZ (Part 1)**
>
> **General Reply**
>
> Thank you very much for your valuable comments. Below, we provide detailed responses to each of your questions and comments. If you have any further questions or concerns regarding our paper or our responses, please let us know, and we will address them promptly.
> ___
> **W1 and Q3: Window Attention**
> > W1: The method section is not clear enough. The mechanisms of window attention and shifting window attention are still not clear.
>
> > Q3: Details for the window attention and shifting window attention are missing. Only Figure 4 shows some fundamental ideas of them but how do they work exactly? How does the shifting window work?
>
> **A1:** Thank you for your helpful suggestions. We will add explanations about Window Attention and Shift Window Attention in the paper and mark them in red for better understanding of the readers. Window attention and shifted window attention, proposed in Swin Transformer [1], introduce a visual inductive bias—similar to the concept of convolution—into the transformer architecture, resulting in a significant performance boost in computer vision tasks. Specifically, window attention computes attention only on tokens within a local spatial window of the image, while shifted window attention applies spatial translation to the window divisions, enabling connections between neighboring tokens across different windows. Since it is not the main contribution of this paper, more details about window attention and shifting window attention can be found in Swin-Transformer [1] and its source code (https://github.com/microsoft/Swin-Transformer/blob/main/models/swin_transformer.py).
>
> [1] Swin Transformer: Hierarchical Vision Transformer using Shifted Windows. ICCV 2021 best paper.
> ___
> **W2 and Q4: The Application of Text-to-MV Task**
> > W2: The task of T2MV is a little strange. What would be an application scenario for that task?
>
> >Q4: Why Text-to-MV task is interesting? At the training level, there is no difference from this task to Text-to-video unless the training data; while for real practice, it's hard to control the generated content from only texts without any reference image.
>
> **A2:** Thanks for your valuable comment. Text-to-MultiView (T2MV) has a wide range of application scenarios such as Virtual Reality (VR), E-commerce, Virtual Shopping, Game Character Design and Fashion Design. Although the output form is the same, the text-to-video task focuses on ensuring motion continuity, while the text-to-multiview task prioritizes the structural and textural consistency of the subject across multiple viewpoints. Additionally, some methods use video generation results as an intermediate step to optimize 3D Gaussian splatting, aiming to improve overall performance.
> ___
>
> **W3:** The "text-to-any-task" is overclaimed. The focused three tasks are essentially video generation (with a single frame or multi-view video).
>
> **A3:** Thank you for your insightful suggestion. Following your recommendation, we will revise 'text-to-any-task' to 'text-to-image/video-task.' We will explore voice and text-based tasks by adapting the 1D proxy token and 1D window attention in the future.
> ___
>
> **Q1:** l201-203, f, h, w are not the actual frame, height, and width of the video/image, but the latent, right? Also, for images, how does the 3D VAE, with temporal downampling layers, handle them?
>
> **A4:** Thanks for your question. Yes, in lines 201-203, f, h, w are the frame, height, and width of latent code at latent space. Current 3D VAEs, such as 3D VAE in Open-Sora, OD-VAE in Open-Sora-PlanV1.2, and MagViT in EasyAnimate, all support single-image inputs by bypassing the temporal down-sampling layers. When processing a video, the temporal down-sampling layers are applied as usual.
> ___
>
> **Q2:** the texts from l242 to l244 are confusing, (pf⋅pf,ph) should be (pf, ph, pw)? so we have ph=pw = <some value depends on resolution> and pf = 1 for image and pf = 4 for videos, right?
>
> **A5:** Thank you for your careful review, and we apologize for the ambiguity caused by our clerical error. Indeed, the text from lines 242 to 244, which currently refers to (pf⋅pf, ph), should be corrected to (pf, ph, pw). We set pf = 1 for images and pf = 4 for videos. We will revise the description in the paper and highlight the changes in red.
>
> ___

---

> ### Author Response · Authors · 2024-11-22
> **Response to Reviewer wrGZ (Part 2)**
>
> **Q5: The performance of T2V-MV**
> > It's hard to evaluate the performance of T2V-MV. Can the T2V-MV work for out-of-distribution prompts? And how about the visual and quantitative evaluations with other models, given that the task setting is following previous approaches?
>
> **A6:** Thank you for your question. Previous work, VideoMV, uses the video output from the text-to-multiview (T2MV) task to assist in optimizing 3D Gaussian Splatting (GS) or Neural Radiance Fields (NeRF), and then renders new views based on the learned 3D GS or NeRF, evaluating the results using PSNR, SSIM, and LPIPS metrics. Since our work focuses solely on the video generation component, a direct comparison with these previous methods is not possible. However, we provide a visual comparison between the results generated in the first stage of VideoMV and those produced by our method in an **anonymous repository**.
> ___
> **Q6:** The page limit is 10, no need to move related work to the appendix.
>
> **A7:** Thank you for your suggestion. Following your advice, we will incorporate the related work into the text.
> ___
> **Q7: A Higher Compression Ratio at 3D VAE and Patch Embedding**
> > Can we use a VAE with a higher compression ratio, or a large patch embedding to achieve a similar level of efficiency? In principle, with the proposed design, more details could be restored compared with directly using a higher compression ratio (but with vanilla attention). It would also be great to show that.
>
> **A8:** Thank you for your insightful suggestions. Using larger compression ratios and larger patch embedding sizes would lead to significant performance degradation due to the loss of too much image detail. Previous experiments and studies on Stable Diffusion and Diffusion Transformers have validated 8x spatial compression and 2x patch embedding compression as more efficient settings that avoid significant performance loss. Therefore, following previous works, our approach does not increase this compression ratio, instead maintaining the token count to preserve the richness of detailed information and prevent performance degradation.
> ___
> **Details Of Ethics Concerns:** The model name "Qihoo-T2X" has nothing to do with the proposed PT-DIT but potentially indicates the identity of the researchers. The authors also did not explain the model name in the paper.
>
> **A9:** Thanks to your suggestion, we will modify the title and paper of Qihoo-T2X to PT-T2I/V to match our proposed PT-DiT.

---

> ### Comment · Area_Chair_j74W · 2024-11-25
> **review the rebuttal**
>
> Dear Reviewer wrGZ,
>
> Could you kindly review the rebuttal thoroughly and let us know whether the authors have adequately addressed the issues raised or if you have any further questions.
>
> Best,
>
> AC of Submission715

---

> > ### Comment · Reviewer_wrGZ · 2024-12-02
> > **Response to rebuttal**
> >
> > Thanks to the authors for the response. My concerns have been addressed. I am still learning towards acceptance thereby maintaining my original rating.

---

### Official Review · Reviewer_g2Xu · 2024-11-04

**Soundness:** 3
**Presentation:** 2
**Contribution:** 3
**Rating:** 6
**Confidence:** 3

**Summary:**

The paper introduces an efficient Proxy-Tokenized Diffusion Transformer (PT-DiT) for accelerating self-attention mechanism, addressing the redundancy in global self-attention mechanisms of diffusion transformers. It develops Qihoo-T2X family for text-to-any-task applications. PT-DiT uses sparse representative token attention, computing an averaging token from each spatial-temporal window as a proxy, reducing computational complexity. It includes window and shift window attention to enhance detail modeling. The Qihoo-T2X family, which includes models for T2I, T2V, and T2MV tasks, shows competitive performance with reduced computational complexity in image and video generation tasks.

**Strengths:**

1. The PT-DiT addresses the redundancy in global self-attention by using sparse representative token attention, reducing computational complexity.

2. The introduction of window and shift window attention mechanisms helps to capture fine details and textures, improving the quality of generated images and videos.

3. The method is designed to be adaptable to both image and video generation tasks without structural adjustments, demonstrating competitive performance compared to original models and great flexibility in application

4. This method is simple and effective.

5. This paper is well written.

**Weaknesses:**

- Insufficent performance comparison to peer methods such as DAM [1] and AgentAttention [2]. While PT-DiT significantly outperforms baselines, its performance relative to peer methods like DAM and AgentAttention is not clearly superior (see Figure 8). Need more comparison.
- **Need further discussion on the relation and differences to previous methods.**
    - For example, DAM **also finds significant attention map redundancy** by analyzing the Divergence Score and proposes a set of Attention Mediator Tokens. These tokens can also **be seen as a kind of extractor** for extracting information from the original latent codes (Queries and Keys).
    - From this perspective, **based on DAM, PT-DiT only points out that redundancy in attention maps remains due to spatial reasons**, and the proposed average operation and sparse representative token attention may be implicitly included by the interaction between the original Queries/Keys and the Attention Mediator Tokens in DAM.
    - Additionally, PT-DiT doesn't analyze the significance of the observed phenomenon across different denoising timesteps, as DAM does.
- From the ablation study in Table 2(a), the results show that without TCM (an auxiliary module), the performance drops significantly, whereas without GIIM, which is the core component proposed in the paper, the performance decline is much less. Does this imply that auxiliary module TCM is more important than core module GIIM?
- A statistical comparison between Qihoo-T2V with and without PT-DiT is necessary for clearer insights into the model's impact.

[1] Pu Y, Xia Z, Guo J, et al. Efficient diffusion transformer with step-wise dynamic attention mediators[J]. arXiv preprint arXiv:2408.05710, 2024.

[2] Han D, Ye T, Han Y, et al. Agent attention: On the integration of softmax and linear attention[J]. arXiv preprint arXiv:2312.08874, 2023.

**Questions:**

See weaknesses.

---

> ### Author Response · Authors · 2024-11-22
> **Response to Reviewer g2Xu (Part 1)**
>
> **General Reply**
>
> Thank you very much for your valuable comments. Below, we provide detailed responses to each of your questions and comments. If you have any further questions or concerns regarding our paper or our responses, please let us know, and we will address them promptly.
> ___
> **W1 and Q1: Performance Comparison with DAM**
> > Insufficent performance comparison to peer methods such as DAM [1] and AgentAttention [2]. While PT-DiT significantly outperforms baselines, its performance relative to peer methods like DAM and AgentAttention is not clearly superior (see Figure 8). Need more comparison.
>
> **A1**: Thank you for your valuable suggestions. Due to an oversight in our comparison with DiT and DAM in Figure 8, we initially used the text-based image generation model for the computational complexity comparison. However, when the same model configuration (e.g., hidden dimensions, number of layers, and attention heads) is applied to class-conditioned image generation, the number of parameters in the test model increases to 192M (due to the substitution of AdaLN-Single for AdaLN), which makes the comparison unfair. To ensure a fair comparison, we have reset the parameter count of PT-DiT/B-Class to 152M, aligning it with the 131M parameters of DAM-B and DiT-B. The results, shown in the following table, demonstrate that our method offers a significant computational complexity advantage over both DiT and DAM. We will update Fig. 8 in the revised pdf.
>
> | Model      | GFLOPs at 256px | GFLOPs at 512px | GFLOPs at 1024px |
> | ----------- | ----------- | ----------- |----------- |
> | DiT-B     | 23.0       |106.5       |650.2       |
> | DAM-B   | 22.5        |90.1       |360.6       |
> | PT-DiT/B-Class   | 18.3        |68.6       |269.6       |
>
> To further compare the performance with DAM, we conducted class-to-image experiments on ImageNet 256. Since DAM is trained on a SiT-based codebase and our training time is limited, we chose PT-DiT/S-Class for a fair comparison with DiT-S, SiT-S, and DAM-S, using two different codebases (DiT-based and SiT-based). The experimental results, shown below, demonstrate that our approach significantly outperforms both DiT-S and SiT-S. This advantage is primarily due to the introduction of window-attention, which aligns with visual priors, as well as the efficiency gains from the proxy token mechanism. For DAM, PT-DiT achieves comparable performance while offering a significant computational complexity advantage (i.e., 3.8 GFLOPs for PT-DiT vs. 5.8 GFLOPs for DAM).
>
> | Model | Codebase | Step | Params(M) | GFLOPs | FID | sFID | IS | Precision | Recall |
> | ----------- | ----------- | ----------- |----------- |----------- |----------- |----------- |----------- |----------- |----------- |
> | DiT-S | DiT | 400k | 33.1 | 6.1 | 68.40 | - | - | - | - |
> | PT-DiT/S-Class   | DiT | 400k | 32.5 | 3.8 | 65.86 | 11.29 | 22.09 | 0.37 | 0.57 |
> | SiT-S | SiT | 400k | 33.1 | 6.1 | 58.61 | 9.25 | 24.21 | 0.41 | 0.59 |
> | DAM-S | SiT | 400k | 33.1 | 5.8 | 53.57 | 9.01 | 27.26 | 0.43 | 0.61 |
> | PT-DiT/S-Class   | SiT | 400k | 32.5 | 3.8 | 55.05 | 9.37 | 26.86 | 0.42 | 0.59 |

---

> ### Author Response · Authors · 2024-11-22
> **Response to Reviewer g2Xu (Part 2)**
>
> **W2: Further discussion**
> > Need further discussion on the relation and differences to previous methods
>
> **A2:**  Thank you for your valuable suggestions, we further discuss the similarities and differences between our approach and DAM below.
>
> **Similarities**: Both PT-DiT and DAM address self-attention redundancy and enhance the efficiency of the diffusion transformer by reducing the number of tokens involved in global interactions.
>
> **Differences**: This paper differs from DAM in the following aspects:
>
> **1.The analysis of redundancy**: We conducted a detailed analysis of the causes and distribution of redundant computations in global self-attention. Visualizing the attention map reveals that redundancy primarily arises from the sparse nature of visual information. Redundant computations mostly occur in the attention interactions between spatially neighboring tokens and distant tokens, while attention between spatially proximate tokens is critical and should not be overlooked.
>
> **2.Proxy token mechanism**: We propose a spatial and temporal prior-based proxy token extraction and modeling, which is a better fit for the visual generation task. DAM may be able to implicitly achieve similar goals by proceeding with training, but it increases the burden of model learning.
>
> **3.Texture complement module**: The interaction between spatially neighboring tokens is preserved through window attention and shifted window attention. In contrast, the mediator token mechanism in DAM may lose some of the spatial relationships between neighboring tokens.
>
> **4.Generic text-to-image/video task**: We focus on building a powerful and foundational text-to-image and video generation model from scratch, whereas DAM is explored solely on the class-to-image task on ImageNet.
>
> To summarize, this paper differs significantly from DAM in terms of redundancy analysis, the proposed methodology, and the primary tasks. We will add a brief discussion in the related work of the revised paper and mark it in red.
>
> > Additionally, PT-DiT doesn't analyze the significance of the observed phenomenon across different denoising timesteps, as DAM does.
>
> **Answer**: Thank you for your insightful suggestions. We believe that designing dynamic compression ratio strategies for PT-DiT, determined by the observed phenomenon across different denoising timesteps, is a valuable research direction. Since each step tends to prioritize different aspects of the generated image, earlier steps may focus on building the overall structure, requiring smaller compression ratios to maintain high-quality global modeling. In contrast, later steps often focus on refining texture details and may benefit from larger window sizes and higher compression ratios for more detailed local modeling. However, adjusting the compression ratio at different steps could potentially lead to training instability, which would require the development of specialized strategies and thorough experimental validation. As a result, we did not analyze this phenomenon across different denoising timesteps or design a dynamic compression ratio strategy in the current paper. However, we plan to explore this idea in future work. Thank you again for your valuable suggestions.
> ___
> **W3: The effectiveness of GIIM and TCM**
> > From the ablation study in Table 2(a), the results show that without TCM (an auxiliary module), the performance drops significantly, whereas without GIIM, which is the core component proposed in the paper, the performance decline is much less. Does this imply that auxiliary module TCM is more important than core module GIIM?
>
> **A3:** Thank you for your question.
> TCM is also a core module in our method. As we pointed out in Section 3.1, the association of spatially neighboring tokens is crucial and cannot be ignored in both visual understanding and generation tasks. While GIIM efficiently achieves global modeling, it cannot model detailed information in images or videos. Since generative tasks are dense prediction tasks, the absence of dense spatial proximity token associations leads to significant performance degradation.
>
> Regarding the observation that the performance degradation of GIIM is less pronounced than that of TCM, we believe there are two main reasons:
>
> 1.GIIM focuses on constructing the overall image layout and ensuring global consistency, a property that is less sensitive to quantitative measures like FID (Fréchet Inception Distance) when compared to the more detailed characteristics captured by TCM.
>
> 2.SWA (Shifted Window Attention) indirectly facilitates the transfer of information across all tokens as the network deepens, thereby supplementing some of the global modeling capabilities.
>
> In summary, GIIM and TCM are not independent modules, but complementary components that work together. Both are essential to the design of PT-DiT, and each plays an irreplaceable role in its effectiveness.

---

> ### Author Response · Authors · 2024-11-22
> **Response to Reviewer g2Xu (Part 3)**
>
> **W4: The Statistical Comparison between w and w/o PT-DiT**
> > A statistical comparison between Qihoo-T2V with and without PT-DiT is necessary for clearer insights into the model's impact.
>
> **A4:** Thanks for your question. We provide a statistical comparison between Qihoo-T2V with and without PT-DiT on UCF-101 and MSR-VTT, as shown in the table below. Due to limited training time, we train the 256×256×16 video generation model on WebVid for 40,000 steps with a batch size of 256. Specifically, we use two commonly adopted video generation architectures: 2D+1D attention and 3D attention. The experimental results show that thanks to the proxy token mechanism and TCM, PT-DiT achieves performance comparable to 3D full attention, with faster convergence due to the introduction of visual priors. In contrast, the 2D+1D approach experiences more significant performance degradation.
> ___
> **UCF-101**
> | Method | FVD | IS| FID |
> | ----------- | ----------- | ----------- |----------- |
> | 3D | 631.02 | 28.36 | 66.37 |
> | 2D + 1D | 679.30 | 25.64 |71.23 |
> | PT-DiT | 598.21 | 29.85 | 64.96 |
> ___
>
> **MSR-VTT**
> | Method | FVD | CLIPSIM|
> | ----------- | ----------- | ----------- |
> | 3D | 438.63 | 0.2295 |
> | 2D + 1D | 509.36| 0.2242 |
> | PT-DiT | 409.34 | 0.2331 |

---

> ### Comment · Area_Chair_j74W · 2024-11-25
> **Reviewer response needed**
>
> Dear Reviewer g2Xu,
>
> Could you kindly review the rebuttal thoroughly and let us know whether the authors have adequately addressed the issues raised or if you have any further questions.
>
> Best,
>
> AC of Submission715

---

> > ### Comment · Reviewer_g2Xu · 2024-11-25
> > **Response to authors' rebuttal**
> >
> > Thank you for the author's response and clarification, which has resolved most of my concerns. I choose to maintain the score.

---

### Official Review · Reviewer_h1hT · 2024-11-04

**Soundness:** 2
**Presentation:** 3
**Contribution:** 2
**Rating:** 6
**Confidence:** 4

**Summary:**

This paper introduces a proxy-tokenized mechanism for transformer-based diffusion models (e.g., DiT), utilizing sparse representative tokens to alleviate redundant computations arising from inherent visual redundancy. Specifically, the authors observe that tokens representing spatially proximate regions tend to exhibit high similarity, while tokens corresponding to spatially distant regions contribute less to meaningful computation in the attention map. Leveraging this insight, the authors propose the use of a proxy token for each window by averaging the tokens within it and subsequently employing self-attention and cross-attention operations involving both proxy tokens and original tokens to ensure efficient interaction across all tokens. Furthermore, the proposed approach integrates a Swin-based architecture to implement efficient attention mechanisms in accordance with these observations. The method achieves competitive performance compared to state-of-the-art approaches across three tasks—text-to-image, text-to-video, and text-to-multiview—at varying resolutions, underscoring its effectiveness and significance within the field.

**Strengths:**

- The paper is well-crafted, providing a clear exposition of the proposed methodology.
- The approach is conceptually straightforward and highly intuitive.
- The motivation and objectives are innovative, focusing on a token-efficient mechanism for DiT-based generative models, which is particularly significant given the paucity of existing approaches for DiT-based models.

**Weaknesses:**

- The lack of ablation studies assessing performance metrics, such as FID with respect to GFLOPs, diminishes the practical relevance of the proposed methodology.
- The discussion and comparison do not sufficiently address related works on token merging (e.g., ToMe [1], ToMeSD [2], CrossGET [3], and TRIPS [4]), which limits the comprehensiveness of the literature review.
- The proposed method integrates a Swin-based architecture within the DiT-based model, employing a straightforward average operation to represent tokens. From the perspective of novelty, this contribution appears to be incremental.



[1] Token Merging: Your ViT But Faster, ICLR 2023

[2] Token Merging for Fast Stable Diffusion, CVPR Workshop 2023

[3] CrossGET: Cross-Guided Ensemble of Tokens for Accelerating Vision-Language Transformers, ICML 2024

[4] TRIPS: Efficient Vision-and-Language Pre-training with Text-Relevant Image Patch Selection, EMNLP 2022

**Questions:**

- Although the authors provide extensive comparisons between resolution and computational metrics (e.g., memory usage, GFLOPs), I believe that additional comparisons between performance and computational cost relative to other approaches in text-to-image and text-to-multiview tasks are necessary. Assessing the trade-off between performance and computational efficiency, rather than focusing solely on resolution, would offer a more comprehensive evaluation of the proposed method's effectiveness. Since identical resolutions do not inherently guarantee equivalent performance, it is conceivable that approximate methods may result in some performance degradation.

- Several prior works, such as ToMe, ToMeSD, TRIPS, and CrossGET, have also addressed token merging. The absence of a detailed discussion or comparison with these existing studies undermines the claimed novelty of Qihoo-T2X. I suggest that the authors more explicitly delineate the unique contributions and advantages of their approach in relation to these established methods.

---

> ### Author Response · Authors · 2024-11-22
> **Response to Reviewer h1hT (Part 1)**
>
> **General Reply**
>
> Thank you very much for your valuable comments. Below, we provide detailed responses to each of your questions and comments. If you have any further questions or concerns regarding our paper or our responses, please let us know, and we will address them promptly.
> ___
> **W1 and Q1: Comprehensive Evaluation of Performance and Efficiency**
> > W1: The lack of ablation studies assessing performance metrics, such as FID with respect to GFLOPs, diminishes the practical relevance of the proposed methodology.
>
> >Q1:Although the authors provide extensive comparisons between resolution and computational metrics (e.g., memory usage, GFLOPs), I believe that additional comparisons between performance and computational cost relative to other approaches in text-to-image and text-to-multiview tasks are necessary. Assessing the trade-off between performance and computational efficiency, rather than focusing solely on resolution, would offer a more comprehensive evaluation of the proposed method's effectiveness. Since identical resolutions do not inherently guarantee equivalent performance, it is conceivable that approximate methods may result in some performance degradation.
>
> **A1**: Thank you for your valuable comment. Due to differences in training data, parameter scale, and training iteration across different text-to-image and text-to-multiview methods, it is challenging to make a perfectly fair comparison of efficiency and performance for text-to-image tasks. As a result, we adopt the widely used class-to-image task on ImageNet 256 to fairly compare PT-DiT with DiT, using the same parameters, training data, and experimental configurations.
> Specifically, we design PT-DiT/L-Class and PT-DiT/S-Class to align with DiT-XL and DiT-S, respectively, based on parameter scale, and conduct experiments using the same training and testing frameworks as DiT. The detailed model settings can be found in Table 4 of the revised paper. The results are shown in the table below, with the data reported without the use of CFG. Compared to DiT, PT-DiT achieves competitive performance and a significant efficiency advantage with the same or fewer steps, thanks to the well-designed proxy token mechanism and texture refinement module. For example, PT-DiT/L-Class at 2000k steps (limited by time constraints) achieved an FID of 9.40, which is lower than the 9.62 FID obtained by DiT-XL at 7000k steps. Similarly, PT-DiT/S-Class at 400k steps achieved an FID of 65.86, outperforming DiT-S, which had an FID of 68.40 at the same step count. Meanwhile, the computational complexity is reduced by 30% (82.64 vs. 118.69 GFLOPs) and 37% (3.86 vs. 6.06 GFLOPs), respectively. These results demonstrate that our method can maintain competitive performance while benefiting from reduced computational complexity.
>
> | Model | Step | Params(M) | GFLOPs | FID | sFID | IS | Precision | Recall |
> | ----------- | ----------- | ----------- |----------- |----------- |----------- |----------- |----------- |----------- |
> | DiT-XL | 7000k | 675 | 118.7 | 9.62 | 6.85 | 121.50 | 0.67 | 0.67 |
> | PT-DiT/L-Class   | 2000k | 693 | 82.6 | 9.40 | 6.17 | 125.07 | 0.68 | 0.66 |
> | DiT-S | 400k | 33.1 | 6.1 | 68.40 | - | - | - | - |
> | PT-DiT/S-Class   | 400k | 32.5 | 3.8 | 65.86 | 11.29 | 22.09 | 0.37 | 0.57 |
>
> Unlike previous token merging approaches, our method preserves all detailed tokens without information loss, while also maintaining the modeling of spatially neighboring tokens—an essential aspect in visual tasks—through Window Attention. As a result, PT-DiT remains competitive and highly efficient, without suffering from noticeable performance degradation.
> ___
>
> **W2: Literature Review**
> > The discussion and comparison do not sufficiently address related works on token merging (e.g., ToMe [1], ToMeSD [2], CrossGET [3], and TRIPS [4]), which limits the comprehensiveness of the literature review.
>
> **A2**: Thank you for your suggestion. We will add the relevant literature in the related work and highlight the changes in the paper.
> ___

---

> ### Author Response · Authors · 2024-11-22
> **Response to Reviewer h1hT (Part 2)**
>
> **W3 and Q2: Novelty of PT-DiT**
> > W3: The proposed method integrates a Swin-based architecture within the DiT-based model, employing a straightforward average operation to represent tokens. From the perspective of novelty, this contribution appears to be incremental.
>
> >Q2: Several prior works, such as ToMe, ToMeSD, TRIPS, and CrossGET, have also addressed token merging. The absence of a detailed discussion or comparison with these existing studies undermines the claimed novelty of Qihoo-T2X. I suggest that the authors more explicitly delineate the unique contributions and advantages of their approach in relation to these established methods.
>
> **A3**: Thank you for your suggestion.
>
> **First and foremost**, the motivation behind our research differs significantly from existing approaches. Our goal is to design an efficient Diffusion Transformer that addresses the challenges of full-attention computational complexity in high-resolution and long-duration text-to-image and text-to-video tasks. In doing so, we aim to develop a series of powerful, generalized text-to-image and text-to-video generators. Among existing approaches, ToMe, TRIPS, and CrossGET are focused on visual and textual understanding, enhancing the models' ability to process both images and language efficiently through token merging. ToMeSD, while applied to generative tasks, merely introduces token merging as an acceleration strategy for existing models.
>
> **Second**, our approach differs significantly from existing token merging methods.
> Simple token merging faces several limitations in generative tasks and cannot effectively build a powerful visual generation model.
>
> 1)	Token merging typically results in the loss of detailed information, whereas generative tasks—particularly those involving sense prediction—place more stringent demands on detail preservation. Excessive loss of detail can lead to significant performance degradation.
>
> 2)	Token merging compromises the ability to model spatially neighboring tokens. By visualizing the attention map, we observe that while long-distance modeling of spatially distant tokens can introduce redundancy, the attention between spatially neighboring tokens is crucial. The token merging strategy, by fusing neighboring tokens, loses part of this spatial relationship, leading to performance degradation.
>
> Based on the above analysis, we do not adopt the token merging strategy. Instead, we design a proxy token mechanism to establish global associations using a limited number of proxy tokens, guided by visual-spatial priors. Meanwhile, all tokens are retained to prevent the loss of detailed information, and the global associations in the proxy tokens are propagated to all detailed tokens. Additionally, the modeling of spatially neighboring tokens is enhanced through the use of window attention and shifted window attention strategies.
>
> Extensive experimental results demonstrate that our approach can build a general and powerful image and video generation model without relying on any pre-training from existing diffusion models. It achieves competitive performance compared to current state-of-the-art image and video generation models, with notable advantages in efficiency and memory usage.
>
> In summary, the unique contributions of our approach can be summarized as follows:
>
> 1.We conduct an in-depth analysis of the redundant computation in self-attention within Diffusion Transformers, caused by visual sparsity and redundancy. We find that redundant computations primarily occur in the attention interactions between tokens within the same spatial window and those that are spatially distant. In contrast, modeling the relationships between spatially adjacent tokens is crucial and should be preserved.
>
> 2.We designed a proxy token mechanism that leverages spatial priors to perform local token fusion and extract proxy tokens while retaining all detail tokens. Proxy token attention is then used to enable global modeling, which is propagated to all dense detail tokens, efficiently establishing global associations without losing local texture details. Additionally, window attention and shifted window attention are introduced to enhance the modeling of spatially adjacent tokens, further improving the capture of spatial proximity details.
>
> 3.Through extensive qualitative and quantitative experiments, we demonstrate that an efficient proxy-token-based diffusion model can achieve competitive performance with existing state-of-the-art text-to-image (T2I) and text-to-video (T2V) models. Our approach builds foundational T2I and T2V models for a range of generalized scenarios without relying on any pre-trained diffusion transformers, offering advantages in computational complexity.
>
> 4.We will open-source our image and video generation models to support the advancement of the efficient diffusion transformer community.
>
> Following your suggestion, we summarize the contributions of PT-DiT in the revised PDF and highlight them in red.

---

> > ### Comment · Reviewer_h1hT · 2024-11-23
> >
> > I appreciate the authors' responses, which have clarified the novelty of this work relative to other token merging approaches, leading me to increase my score. Nevertheless, some concerns persist. Specifically, the current version of the paper does not adequately convey the distinct efficiency advantages of Qihoo-T2X.
> >
> > As stated in my initial review, the comparisons regarding performance and efficiency remain insufficient. While the current version provides some similar results, such as those in Table 1(b), I believe that additional comparative visualizations would be highly beneficial. For example, Figures 8 and 9 could be revised to include axes representing metrics like FID (or IS) and parameters (or compression ratio). The results presented in Table 2(d) demonstrate that different compression ratios yield differing levels of performance. Therefore, incorporating comparative figures across multiple tasks would effectively highlight the efficiency of the proposed method in contrast to other approaches. This would facilitate demonstrating scenarios where Qihoo-T2X achieves superior performance at specific compression ratios or matches the performance of alternative methods while requiring fewer parameters.
> >
> > In sum, providing a more comprehensive discussion of related work would help readers more thoroughly understand the distinctive contributions and merits of the proposed approach.

---

> > > ### Author Response · Authors · 2024-11-26
> > > **Response to Reviewer h1hT**
> > >
> > > Thank you for your meticulous review and your kind compliments on our response!
> > >
> > > We have updated the revised paper and highlighted the changes in red. In response to your suggestion, we have added a discussion of existing methods in the "Related Work" section, along with a comprehensive comparison of performance and efficiency, including axes representing FID metrics (see Appendix 2). While conducting further comparisons of performance and efficiency is challenging due to high training and time costs, we plan to explore this aspect in more detail in future studies.
> > >
> > > Once again, thank you for your valuable feedback and contributions to improving this article!

---

### Official Review · Reviewer_vCtb · 2024-11-08

**Soundness:** 3
**Presentation:** 3
**Contribution:** 3
**Rating:** 6
**Confidence:** 3

**Summary:**

This paper proposes Proxy-Tokenized Diffusion Transformer to reduce the computational complexity. By utilizing sparse representative token attention, window and shift window attention, the proposed method effectively reduces redundancy in global self-attention computation and optimizes the modeling process of visual information. Also, the Qihoo-T2X family includes multiple models for text to image, text to video, and text to multi view tasks. However, it is difficult to fully demonstrate the effectiveness of the method at various resolutions in the experimental section.

**Strengths:**

The proposed method significantly reduces computational complexity and has certain competitiveness in various tasks, such as T2I, T2V,and T2MV.

**Weaknesses:**

1)	Insufficient experiments and lack of comparative experiments at high resolution.
2)     How is the value of compression ratio determined, and is it related to both semantic information and resolution size?

**Questions:**

See weakness.

---

> ### Author Response · Authors · 2024-11-22
> **Response to Reviewer vCtb**
>
> **General Reply**
>
> Thank you very much for your valuable comments. Below, we provide detailed responses to each of your questions and comments. If you have any further questions or concerns regarding our paper or our responses, please let us know, and we will address them promptly.
> ___
> **W1: High Resolution Results**
> > Insufficient experiments and lack of comparative experiments at high resolution.
>
> **A1**: Thanks for your valuable suggestion. We apologize for the omission of the image resolution in the text-to-image section of the Qualitative Analysis (Sec. 4.2). To clarify, Fig. 5 presents the results of a qualitative comparison with an existing state-of-the-art model at a high resolution of 1024px, where our method demonstrates competitive performance. We will revise the paper to include this information, which will be highlighted in red. In addition, higher-resolution results (1024px * 2048px, 2048px * 1024px, 1792px * 1792px) are provided at the **Anonymous Repository**.
> ___
> **W2: The Value of Compression Ratio**
> > How is the value of compression ratio determined, and is it related to both semantic information and resolution size.
>
> **A2**: Thank you for your suggestion. **We first determine the compression ratio at a resolution of 256.** At this resolution, after applying the VAE (8x down-sampling) and patch embedding (2x down-sampling), the image is reduced to only 16x16 tokens. With a compression ratio of (1, 4, 4), the number of windows in the space becomes 4x4, and the number of proxy tokens is 16. We found that, with this configuration, the limited number of windows and the large coverage area of each window make it challenging for a single proxy token to effectively represent the complex information within the window for global information modeling. This leads to anomalies in the image layout and performance degradation, as shown in the Table. 2(d) in the paper. Setting the compression ratio to (1, 2, 2) maintains a reasonable number of windows while preserving the necessary semantic richness within each window. Through experimentation, we found that this setting strikes a balance between maintaining performance and improving efficiency. In contrast, the (1, 1, 1) setting does not support proxy token compression or window-attention, which are crucial for reducing computational complexity. As a result, we chose a compression ratio of (1, 2, 2) for the 256 resolution. **For higher resolutions**, we found that maintaining the same number of windows across different resolutions benefits the model's training process, as it ensures a consistent semantic hierarchy across resolutions. We will revise the paper to clarify this section and highlight the changes in red.

---

> ### Comment · Area_Chair_j74W · 2024-11-25
> **Reviewer response needed**
>
> Dear Reviewer vCtb,
>
> Kindly review the rebuttal thoroughly and let us know whether the authors have adequately addressed the issues raised or if you have any further questions.
>
> Best,
> AC of Submission715

---

### Author Response · Authors · 2024-11-26
**Official Comment by Authors**

Dear reviewers,

We sincerely thank the reviewers for their thoughtful and thorough review, as well as for their praise regarding the clarity of our writing (**h1Ht**, **g2Xu**, **wrGZ**), the motivation behind our study (**h1Ht**, **g2Xu**, **q44Z**), the innovativeness of the proposed methodology (**h1Ht**, **g2Xu**, **q44Z**), and the diversity of our experiments and analyses (**vCtb**, **g2Xu**, **wrGZ**, **q44Z**).
_____
We would also like to express our sincere gratitude for taking the time to review our paper and for offering constructive and valuable feedback to help improve it. We have updated the manuscript to address the points you raised, with the revised sections highlighted in red. Below is a summary of the major revisions:

1.	Add a brief discussion of existing methods in related work.

2.	Modify name of the T2I and T2V models and title of the paper.

3.	Add a description of this paper's contribution to introduction.

4.	Add explanations about window-attention and shift window attention in Sec. 3.2.2 (Texture Complement Module).

5.	Add the process of determining the compression ratio in Sec. 3.2.3 (Compression Ratios).

6.	Correct data in Fig. 8 (now Fig. 10).

7.	Add a comprehensive integrated comparison of performance and efficiency in Appendix. 2.

8.	Fix some writing errors and revise the paper layout.

Authors of Submission715

---

### Meta-Review · Area_Chair_j74W · 2024-12-17

**Metareview:**

(a) The paper proposes the Proxy-Tokenized Diffusion Transformer to reduce computational complexity by using sparse representative token attention and window-based attention. The Qihoo-T2X family includes models for text-to-image, text-to-video, and text-to-multi-view tasks.

(b) The strengths of the paper are: The paper effectively reduces training and inference costs of DiT by minimizing redundant computation through the promising GIIM and TCM designs, demonstrating strong performance across tasks (Text-to-Image, Text-to-Video, Text-to-MultiView) with comprehensive experiments, clear visualizations, and lower data and computational requirements compared to SOTA methods.

(c) The weaknesses of the paper are: unclear method part, overclaimed "text-to-any-task", lack of ImageNet comparisons, hyper-parameter ablation, the mechanism of proxy tokens, lack of related works on token merging and lack of comparative experiments at high resolution.

(d) The most important reasons for acceptance are that this work introduces the sparse representative token attention to model global visual information efficiently, whilst makes solid contributions by introducing a variety of models for a series of  T2I, T2V, and T2MV tasks.

**Additional Comments On Reviewer Discussion:**

(a) Reviewer vCtb shows that insufficient high-resolution experiments hinder analysis, and the compression ratio depends on both semantic information and resolution size. The reviewers have successfully response to these issues.

(b) Reviewer h1hT identifies the lack of ablation studies, insufficient comparisons with related token merging works, and the incremental novelty of integrating a Swin-based architecture with a basic token averaging operation reduce the method's practical relevance and originality. During the rebuttal, the reviewers have successfully clarified the novelty of this work relative to other token merging approaches, leading Reviewer h1hT to increase the score.

(c) Reviewer g2Xu finds that PT-DiT lacks sufficient comparison with peer methods, underexplores differences from prior works like DAM, and raises questions about its core contributions and impact based on ablation and statistical analyses. The authors have resolved most of the concerns and the Reviewer choose to maintain the score.

(d) Reviewer wrGZ shows that the method section lacks clarity on window attention mechanisms, the T2MV task's application scenario is unclear, and the "text-to-any-task" claim is overstated, as it primarily focuses on video generation tasks. The authors actively respond and follow the reviewer's recommendation to revise 'text-to-any-task' to 'text-to-image/video-task.' The reviewer is learning towards acceptance thereby maintaining the original rating.

(e) Reviewer q44Z finds that the paper lacks ImageNet evaluations for fair comparison with diffusion transformer models, offers limited justification for chosen compression ratios, and should explore alternative proxy token extraction methods for improved representativeness. The authors' response have resolved all the questions, and the reviewer suggests that the relevant discussions can be included in the final version of the paper.

---

### Decision · Program_Chairs · 2025-01-22

Accept (Poster)